# Cell Marker Accordion: interpretable single-cell and spatial omics annotation in health and disease

Emma Busarello [1,10] ✉, Giulia Biancon [2,3,10], Ilaria Cimignolo [1], Fabio Lauria [4], Zuhairia Ibnat[1], Christian Ramirez [1], Gabriele Tomè [1,4], Marianna Ciuffreda [1], Giorgia Bucciarelli [1], Alessandro Pilli [1], Stefano Maria Marino [1], Vittorio Bontempi [5], Federica Ress [6], Kristin R. Aass [7], Jennifer VanOudenhove [2], Luca Tiberi [6], Maria Caterina Mione [5], Therese Standal [7], Paolo Macchi [8], Gabriella Viero [4], Stephanie Halene [2,9] ✉ & Toma Tebaldi [1,2] ✉

Single-cell technologies offer a unique opportunity to explore cellular heterogeneity in health and disease. However, reliable identification of cell types and states represents a bottleneck. Available databases and analysis tools employ dissimilar markers, leading to inconsistent annotations and poor interpretability. Furthermore, current tools focus mostly on physiological cell types, limiting their applicability to disease. We present the Cell Marker Accordion, a user-friendly platform providing automatic annotation and unmatched biological interpretation of single-cell populations, based on consistency weighted markers. We validate our approach on multiple single-cell and spatial datasets from different human and murine tissues, improving annotation accuracy in all cases. Moreover, we show that the Cell Marker Accordion can identify disease-critical cells and pathological processes, extracting potential biomarkers in a wide variety of disease contexts. The breadth of these applications elevates the Cell Marker Accordion as a fast, flexible, faithful and standardized tool to annotate and interpret single-cell and spatial populations in studying physiology and disease.

Single-cell RNA sequencing (scRNA-seq) characterizes the transcriptome of each cell in large populations. This high-throughput approach is the ideal choice to reveal the heterogeneous landscape of normal and aberrant cell differentiation processes. It enables the study of cells with diverse properties, including varying self-renewal capacity, multipotent potential, and high plasticity, which are critical in infections, immune responses, and disease pathogenesis[1–3]. Spatial omics further contribute by adding architectural context, revealing how cells are organized within tissues and how this influences their function[4,5].

With the enormous opportunities offered by single-cell technologies, a new set of challenges is rapidly emerging in data analysis and interpretation. Accurate and reliable annotation of cell types is key to deriving faithful biological conclusions. Robustness in identifying cell types is an essential prerequisite to discern disease-critical cells, characterized by aberrant cell states responsible for disease initiation, progression and therapy resistance[6]. In addition, measuring the single-cell activity of gene signatures or modules associated with pathologically relevant pathways is fundamental to unraveling pathogenic mechanisms in aberrant cells[7] and discovering potential disease biomarkers[8].

Identification of cell populations within single-cell data can be executed manually or automatically[9]. Manual annotation, based on the investigator's knowledge or derived from published literature, is generally subjective and often non-reproducible due to a lack of standardization. Many computational tools perform automatic annotation by correlating reference expression data or transferring labels from other single-cell datasets[10–13]. These approaches require reliable transcriptome profiles of purified cells or high-quality annotated single-cell data[14]. However, such reference datasets are not readily available, especially for pathological samples; they can lack the cell populations of interest and might be susceptible to technical specificities such as platform or sequencing strategy[15]. Alternatively, automatic annotation can be achieved by employing predefined sets of cell marker genes[16–18]. Many current tools require the user to provide a collection of markers, a process prone to bias[12,17,19].

We show that currently available gene marker databases are extremely heterogeneous, contain different marker sets for the same cell type, and employ a non-standard nomenclature and classification, thus leading to inconsistent annotation of cell populations in scRNA-seq and spatial data, and poor interpretability of results. Furthermore, current tools and resources focus mostly on physiological cell types, limiting the identification of disease-critical cells. To address these issues and improve the interpretation of normal and aberrant cell types in single-cell and spatial data, we developed the Cell Marker Accordion, an easily accessible and well-documented platform constituted by an interactive R Shiny web application requiring no programming skills and an R package.

The Cell Marker Accordion database is built upon multiple published databases of human and mouse gene markers for cell types (Supplementary Data 1) (both general and tissue-specific), standard collections of widely used cell sorting markers and literature-based marker genes associated with disease-critical cells in multiple pathologies, including liquid and solid tumors. The Accordion database allows marker genes to be weighted not only by their specificity but also by their evidence consistency score, measuring the agreement of different annotation sources. The Cell Marker Accordion web interface permits to explore the integrated collection of marker genes and to easily browse hierarchies of cell types following the Cell Ontology structure to obtain the desired level of resolution with tissue specificity.

The Cell Marker Accordion R package allows to automatically annotate healthy and aberrant populations in single-cell datasets from multiple tissues, exploiting positive and negative markers from either the built-in Accordion database or any gene signature of interest provided by the user. Genes, cell types or pathways that mostly influence annotation results can be easily accessed and visualized to allow the transparent interpretation of results.

We benchmarked the Cell Marker Accordion on multiple single-cell and spatial omics datasets, using surface markers and expert-based annotation as the ground truth. In all cases, we significantly improved the annotation accuracy with respect to existing annotation tools. Moreover, we show that the Cell Marker Accordion can be used to identify pathological processes and disease-critical cells: leukemia cell subtypes, including therapy-resistant cells, in acute myeloid leukemia patients[20,21]; neoplastic plasma cells in multiple myeloma samples[22,23]; malignant subpopulations in glioblastoma and lung adenocarcinoma patients[24,25]; cell type alterations driven by pathologically relevant mutations in myelodysplastic syndromes[26,27]; activation of innate immunity pathways in bone marrow from mice with Mettl3 deletion or treated with METTL3 inhibitors[28,29].

The Cell Marker Accordion is a fast, user-friendly, flexible and comprehensive tool that improves the annotation and interpretation of both physiological and pathological cell populations with single-cell resolution.

## Results

### Widespread heterogeneity across annotation sources leads to inconsistent cell type annotation

To unravel information discrepancies across currently available gene marker databases, we automatically annotated a published scRNA-seq dataset of human bone marrow[30], extracting marker genes from CellMarker2.0[31] and Panglao DB[32], two of the most comprehensive databases for cell type markers (Fig. 1A, see "Methods"). Cell type annotation was often inconsistent between the two sources, showing divergent cell types assigned to the same cluster (for example, "hematopoietic progenitor cell" and "anterior pituitary gland cell") or different nomenclature (for example, "Natural killer cell" and "NK cells"). We unfolded very high discrepancies in marker genes utilized by these resources, with a maximum Jaccard similarity index of 0.23 between matching cell types (Fig. 1B).

To extend this initial observation, we systematically explored the heterogeneity of seven available marker gene databases over common cell types[31–37] (see "Methods"). The comparison showed low consistency between databases, with an average Jaccard similarity index of 0.08 and a maximum of 0.13 (Fig. 1C). These results show that different marker gene databases inevitably lead to inconsistent interpretations of the biological meaning of single-cell data and raise concerns with profound consequences for data mining.

### The Cell Marker Accordion: a user-friendly platform for the annotation and interpretation of single-cell populations

To address the need of robust and reproducible identification of cell types in single-cell datasets, we developed the Cell Marker Accordion, comprising a gene marker database, an R shiny web app and an R package to automatically annotate and interpret single-cell populations.

We built the Cell Marker Accordion database by integrating 23 marker gene databases and cell sorting marker sources (Supplementary Data 1), distinguishing positive from negative markers (Fig. 2A). Standardization was achieved by mapping the initial cell type nomenclature to the Cell Ontology terms[38] and tissue names to the Uber-anatomy ontology (Uberon) terms[39]. Next, via database integration, we obtained a comprehensive set of cell-type-specific marker genes, human and murine, in hundreds of tissues. Importantly, in the Cell Marker Accordion database, genes are weighted by their specificity score (SPs), indicating whether a gene is a marker for different cell types, and by their evidence consistency score (ECs), measuring the agreement of different annotation sources (see "Methods").

The intuitive and interactive Accordion Shiny web interface permits easy retrieval of marker genes associated with input cell types and vice versa, starting from a list of candidate genes to obtain the matching cell types (Fig. 2B right). Hierarchies of cell types can be easily browsed following the Cell Ontology structure to select and obtain the desired level of resolution in the markers. Users can upload a custom sets of genes to either update the repository or perform cell type marker enrichment analysis, with no need for programming skills.

Finally, the Cell Marker Accordion R package allows to automatically annotate cell populations based on the built-in database, with the considerable advantage of weighting the markers according to their evidence consistency and specificity scores (Fig. 2B, left and Supplementary Fig. 1). The automatic annotation can be easily integrated into a Seurat analysis workflow[34], requiring as input only the count matrix or a Seurat object. Built-in lists of positive and negative cell cycle markers can be used to assign the appropriate cell cycle phase to each cell or to evaluate quiescence. Any annotation procedure can be easily enhanced by including custom gene lists associated with cell types, specific pathways or signatures of interest. Importantly, with respect to other tools, the Cell Marker Accordion implements novel options to explore annotation results by providing the top

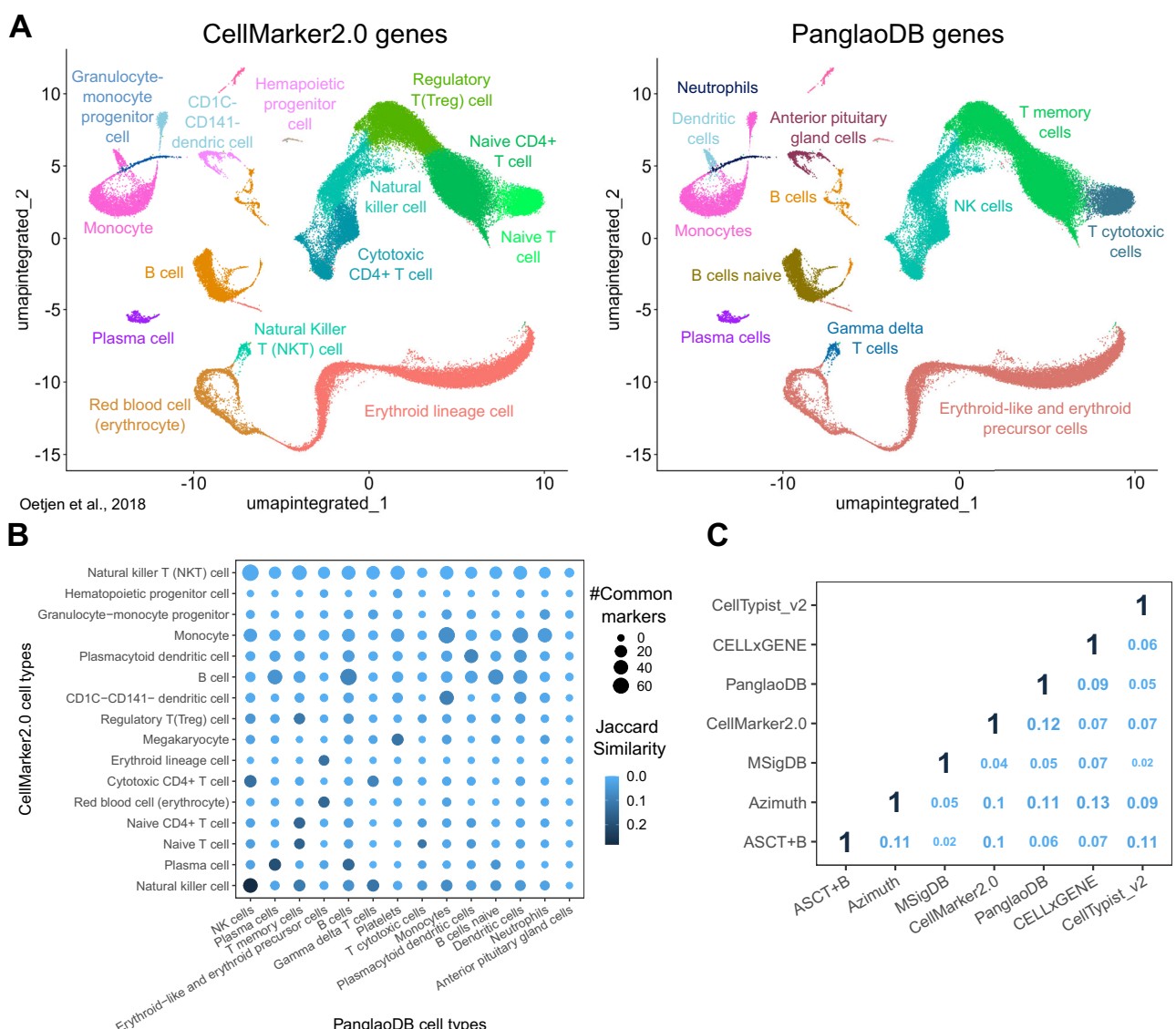

**Fig. 1 | Heterogeneity in marker gene databases leads to inconsistent single-cell annotations. A** Cell type identification by automatic annotation with ScType[40] in a published bone marrow dataset[30], using markers from CellMarker2.0 (left) and PanglaoDB (right) as input. **B** Overlap between marker genes from CellMarker2.0 (y-axis) and PanglaoDB (x-axis). The dot color represents the Jaccard similarity index, and the dot size indicates the number of common markers in each cell type pair. **C** Comparison of cell type markers among seven published databases. The numbers indicate the average Jaccard similarity index between each database pair, calculated using all common cell types. Source data are provided as a Source Data file.

marker genes that most significantly determine the final annotation. The diversity or similarity of the top cell types competing for the same annotation can be evaluated by inspecting their position along the Cell Ontology tree (Supplementary Fig. 2).

### The Cell Marker Accordion improves the annotation of cell types in complex single-cell and spatial multi-omics

To validate the Cell Marker Accordion, we undertook a benchmark study to compare its annotation performance against five other automatic tools based on markers: ScType[40], SCINA[19], clustifyR[12], scCATCH[41] and scSorter[17] (described in Supplementary Data 2), using multiple published human and murine datasets across different tissues and platforms (Fig. 3, Supplementary Figs. 3 and 4).

First, we exploited a 93456-cell single-cell RNA-seq dataset[42] acquired from fluorescent antibody-sorted (FACS) blood cells, based on 15 cell surface markers and resulting in 10 different populations, separately profiled, that we used as ground truth (Fig. 3A). Compared to all the other tools[12,17,19,40,41] (see "Methods"), the Cell Marker

Accordion shows improved cell type assignment (Fig. 3B) and lower running time (Fig. 3C), making it suitable for larger datasets and real-world applications. Furthermore, we provide unique visualizations to boost the interpretation of results in terms of cell types competing for the final annotation (Fig. 3D), together with their similarity based on the Cell Ontology hierarchy (Fig. 3E) and the top influential marker genes (Fig. 3F).

To extend our benchmark, we selected two human bone marrow datasets, obtained via similar multi-omics methods (CITE-seq and AbSeq), that simultaneously captured RNA and protein expressions[43,44]. In the first case, 25 barcoded antibodies were used to quantify surface proteins and identify 14 different cell types in 30,602 total cells (Supplementary Fig. 3A). The second dataset comprised 13,159 cells, classified into 24 different cell types according to the expression of 97 barcoded antibodies (Supplementary Fig. 3B). We considered surface markers as the ground truth to evaluate and compare annotation results. Additionally, we analyzed five single-cell RNA-seq datasets derived from different tissues: immune system

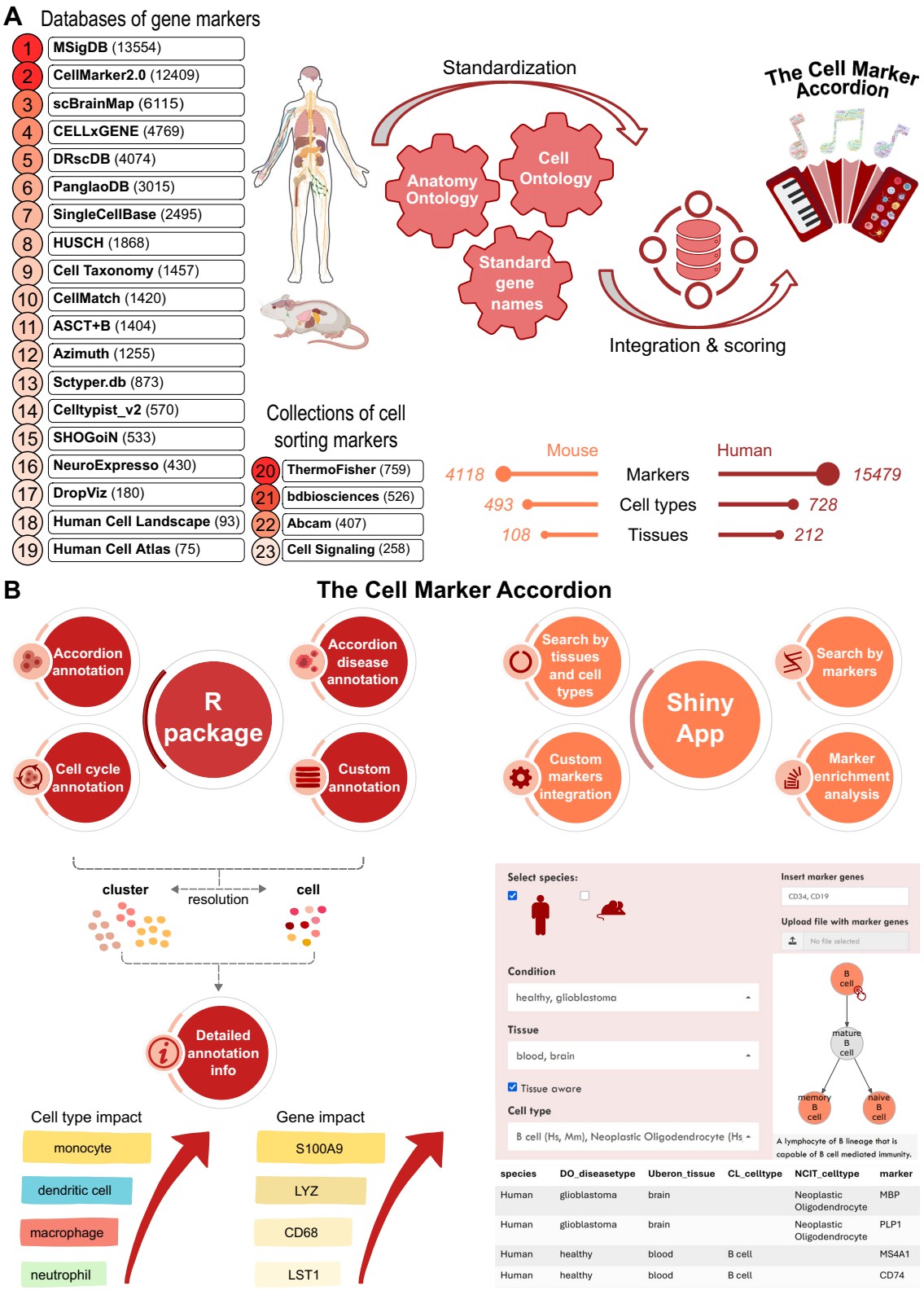

**Fig. 2 | The Cell Marker Accordion: a user-friendly platform for annotating and interpreting single-cell populations. A** Workflow for building the Cell Marker Accordion database. Sources are ranked according to their initial number of markers. The resulting numbers of human and murine markers, cell types and tissues are reported. Mouse and human illustrations created in BioRender. Tebaldi, T. (2025) https://BioRender.com/x09w717. **B** Overview of the main functionalities of the Cell Marker Accordion R package and Shiny app.

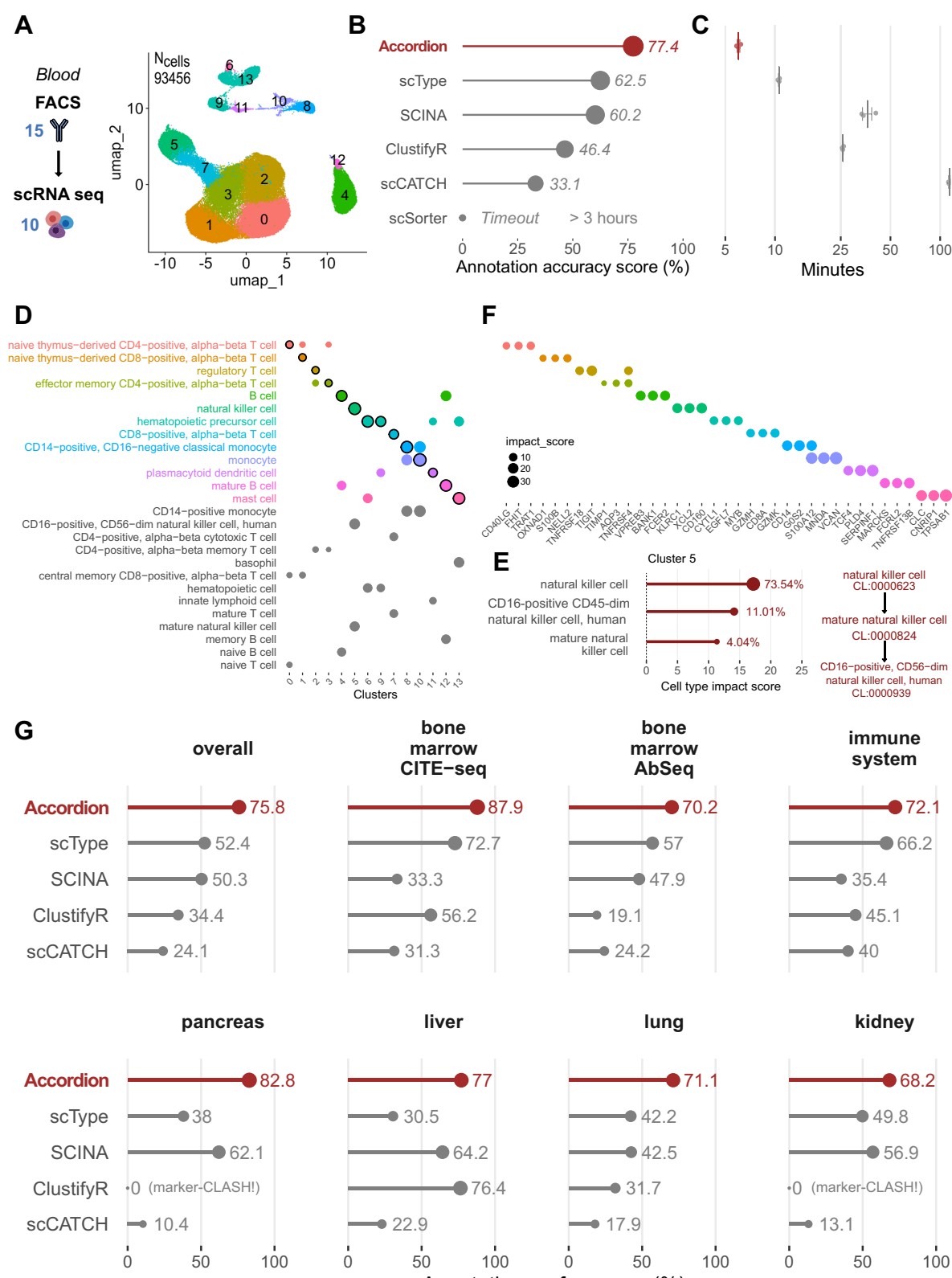

(https://www.ebi.ac.uk/gxa/sc/experiments/E-HCAD-4), pancreas[45], liver[46], lung[47] and kidney[45] all of which featured expert-curated cell type annotations serving as the ground truth (Supplementary Fig. 3C and Supplementary Fig. 4A–D). Notably, in all cases, the Cell Marker Accordion showed improved annotation performances, with an average increase of approximately 23% with respect to the other tools

(Fig. 3G, Supplementary Data 3 for a breakdown of the results in each cell type).

Finally, to show the potential of the Cell Marker Accordion in annotating spatially resolved transcriptomics, we took advantage of a recently published adult mouse brain MERFISH dataset, with a panel of 1122 genes[48]. We considered a single brain coronal section, containing

**Fig. 3 | The Cell Marker Accordion improves the annotation of cell types in multiple tissues from complex single-cell multiomics.** Annotation of single-cell datasets and interpretation of the results with the Cell Marker Accordion, and performance comparison with other marker-based annotation tools. **A** Dataset of PBMC FACS sorted cells separately profiled with single-cell RNA-seq. In total 15 surface antibodies were used to sort 10 different cell types, used as the ground truth. Populations identified by the Accordion are color-coded in the UMAP, with cluster numbers. **B** The Cell Marker Accordion annotation performance, measured as the similarity between the identified cell types and the ground truth (see "Methods"), is compared against other annotation tools. **C** Comparison of running

times across annotation tools (time axis is log scaled). Data are presented as mean values ± SEM (n = 3). **D** Cell Marker Accordion interpretation of results: top three cell types achieving the highest impact score for each cell cluster (the winning cell type is highlighted). **E** Cell type annotation for cluster 5. Left: top three cell types, ordered according to their impact score, with corresponding percentages of cells in the cluster. Right: Cell Ontology tree of the top three cell types. **F** Top three marker genes with the highest impact score for each cell type, color-coded as (**D**). **G** Comparison of annotation performances between the Cell Marker Accordion and other tools in multiple single-cell datasets from different tissues. Source data are provided as a Source Data file.

37,068 cells and 34 annotated cell types, used as ground truth (Fig. 4A and Supplementary Fig. 5). First, we performed cell type annotation with the Cell Marker Accordion (Fig. 4B). Notably, the spatial mapping of the identified cell types revealed high similarity with the original annotation (Fig. 4C). Also in the spatial scenario, the Cell Marker Accordion annotation performance improved by over 13% compared to the other tools (Fig. 4D, see "Methods" and Supplementary Data 3).

Overall, these benchmarking results highlight the Cell Marker Accordion as a novel tool to obtain fast, more robust, consistent and highly interpretable annotation of cell populations in single-cell and spatial data.

## The Cell Marker Accordion identifies disease-critical cells in human pathologies

Many pathologies, including tumors, contain critical cell populations exhibiting altered states and aberrant gene expression. These disease-critical cells significantly influence disease progression and treatment outcomes[20]. The persistence of a selective subset of malignant cells has been considered the underlying cause of the high relapse rates commonly observed in a variety of cancer patients[49–51]. Identifying and characterizing disease-critical cells in human pathologies is pivotal for improving diagnosis towards interceptive medicine[8], understanding pathogenesis and therapy resistance mechanisms, and developing novel therapies to specifically target and eradicate cancer-initiating cells while minimizing adverse effects on healthy cells. To expand the Cell Marker Accordion to the analysis of pathologies, we created a "disease" collection. We integrated marker genes associated with disease-critical cells found in multiple pathologies, including different tumor types affecting blood, brain, lung, pancreas and other tissues (Fig. 5A). To obtain a standardized and consistent vocabulary of pathologies, we mapped disease terms to the Disease Ontology[52].

Notably, we collected more than 1073 markers associated with acute myeloid leukemia (AML), one of the most common types of blood cancer in adults[53]. A major challenge in AML treatment is the survival of a few therapy-resistant cells. Among these, leukemic hematopoietic stem cells (LHSCs) are key factors contributing to disease progression and relapse[20,49,54,55]. Therapy-resistant cells can show neoplastic monocytic features, associated with lower remission rates and overall survival[56–58]. To show the potential of the Cell Marker Accordion in identifying disease-critical cell types, we analyzed a published scRNA-seq dataset of CD34+ bone marrow cells from 5 healthy controls and 14 AML patients[59]. First, healthy cell types were annotated (Fig. 5B). Next, by exploiting LHSC and neoplastic monocyte marker genes, the Cell Marker Accordion was able to assign an LHSC score and a neoplastic monocyte score for each cell (Fig. 5C, D). Notably, we observed an accumulation of LHSCs in AML progenitor populations (Fig. 5C), as well as an increase of neoplastic monocytes within AML monocyte-derived populations (Fig. 5D). To extend our analysis to the context of therapies, we took advantage of another published scRNA-seq dataset of human bone marrow, with sequential samples at diagnosis and relapse from patients treated with the BCL-2 inhibitor venetoclax[57]. As for the previous dataset, we first annotated healthy cell types (Fig. 5E). LHSCs and neoplastic monocytes were identified (Fig. 5F, G). Consistent with published results[27], we detected cells with high LHSC scores within

progenitor populations and high neoplastic monocyte cell scores in monocyte-derived cells at diagnosis (Fig. 5F, G). At relapse, we observed the disappearance of LHSCs, implying that venetoclax-based treatment can target and eradicate them (Fig. 5F, H). Instead, neoplastic monocytes persisted after therapy, confirming, as previously proposed[57,60–62], that the mechanism of resistance to venetoclax resides in an aberrant monocytic population (Fig. 5G, H). These results suggest that malignant cell heterogeneity plays a significant role in treatment response and disease progression[20,61]. To further characterize the properties of leukemia aberrant cell types in AML patients, we extracted with the Cell Marker Accordion the top altered markers defining either LHSCs or neoplastic monocytes (Fig. 5I).

We further demonstrated the potential of the Cell Marker Accordion in identifying disease-critical cells in single-cell datasets from patients with multiple myeloma (MM) (Supplementary Fig. 6). We exploited a published scRNA-seq dataset of sorted bone marrow plasma cells from 11 healthy controls and 12 MM patients[63] (Supplementary Fig. 6A). We identified neoplastic plasma cells with a significantly higher score in MM patients (Supplementary Fig. 6B, C). These cells were clustered in patient-specific groups by the expression levels of specific immunoglobulin variable regions, suggesting distinct clonotypes (Supplementary Fig. 6D–G). Importantly, the Cell Marker Accordion was able to extract genes that are most critical for defining the identity of malignant plasma cells. (Supplementary Fig. 6F).

We next compared the Cell Marker Accordion performance in identifying aberrant tumor populations with respect to two available tools, SCEVAN[64] and CopyKAT[65], which identify tumor cells through calling copy number variations (CNV). For our benchmark, we exploited two published single-cell RNAseq datasets of glioblastoma and lung adenocarcinoma patients[24,25]. In the glioblastoma dataset, malignant cells were identified based on the expression of a gene list provided by the original authors (see "Methods" and Supplementary Fig. 7) and were subsequently used as the ground truth. The lung adenocarcinoma dataset already included a classification of tumoral cells, which was directly used as ground truth. Performance annotation was assessed using two metrics: the percentage of correctly annotated cells and the corresponding F1 scores (see "Methods"). The Cell Marker Accordion achieved the best performance, identifying 100% of the malignant cells in the glioblastoma patients, with an F1 score of 0.97 (Fig. 6A, B). Additionally, the Cell Marker Accordion significantly outperformed the other tools in terms of speed (Fig. 6C). While SCEVAN required more than 6 h and CopyKAT more than 7 days to complete the annotation task, the Cell Marker Accordion achieved this in less than 2 min (Fig. 6C; see "Methods"). Furthermore, while other tools can only distinguish malignant versus non-malignant cells based on inferred CNV signatures, a defining advantage of the Cell Marker Accordion is its ability to dissect aberrant sub-populations based on expression alterations. In the lung adenocarcinoma dataset, the Cell Marker Accordion not only improved the accuracy in classifying malignant cells with respect to SCEVAN and CopyKAT (0.84 vs 0.52 and 0.52 F1 scores, respectively), but was also the only tool able to identify a small subpopulation of endothelial cells with a neoplastic gene expression signature, in agreement with the original publication[25] (Fig. 6D, E).

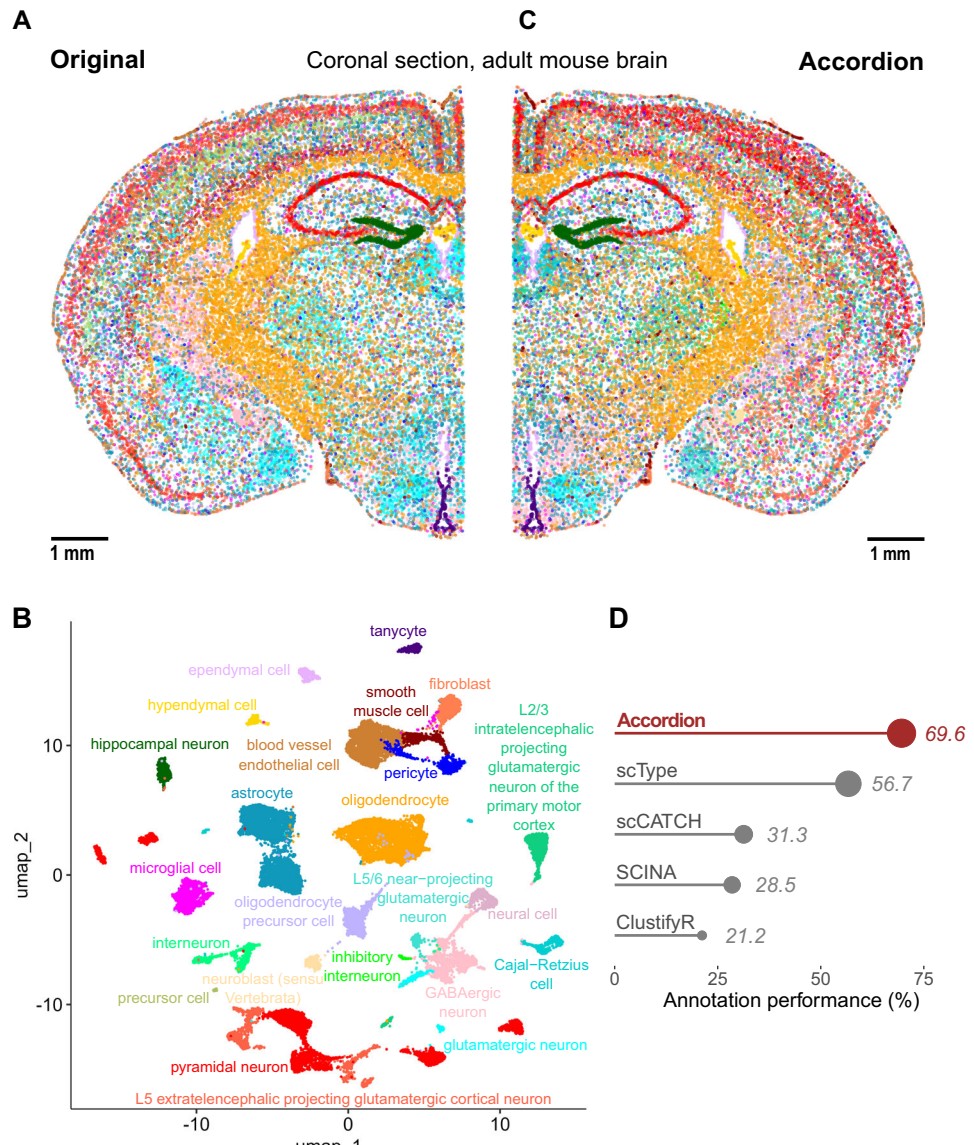

**Fig. 4 | The Cell Marker Accordion improves the annotation of brain cell types in spatial transcriptomics. A** Spatial map and original annotation of a coronal section of an adult mouse brain, analyzed by MERFISH, based on a panel of 1122 genes[48]. Each dot corresponds to a cell, colored by cell type. Scale bar: 1 mm. **B** UMAP plot based on the transcriptional profile of each cell, with colors based on the annotation of the Cell Marker Accordion. **C** Spatial map with cells colored according to cell types as annotated by the Cell Marker Accordion. Scale bar: 1 mm. **D** Comparison across tools of annotation performances, measured as the similarity between predictions and ground truth. Source data are provided as a Source Data file.

Overall, these results provide evidence for the potential of the Cell Marker Accordion to identify malignant and neoplastic cells with aberrant states in human pathologies and to investigate disease mechanisms by extracting altered gene signatures in the quest for biomarker discovery.

## The Cell Marker Accordion identifies altered cell type composition in patients with splicing factor mutant myelodysplastic syndromes

Mutations in splicing factor (SF) genes are prevalent in approximately 50% of patients with myelodysplasia (MDS) and acute myeloid leukemia (AML)[66–68]. These mutations, especially the U2AF1 mutations, are linked to a high risk of AML transformation and decreased survival rates[69–75]. To explore the molecular mechanisms and biological implications that drive the clonal advantage of SF mutant cells over their wild-type counterparts, we conducted single-cell RNA sequencing on CD34+ cells from MDS patients, either without SF mutations ($n = 5$) or

with the U2AF1 S34F mutation ($n = 3$). From a total of 62496 high-quality cells (see "Methods"), we identified cell types with the Cell Marker Accordion (Fig. 7A). Most resulting cell types were related to blood progenitor cells, in line with the CD34+ cell sorting. To investigate the impact of the U2AF1 S34F splicing factor mutation, we compared cell type composition between U2AF1 WT and mutant patients (Fig. 7B). Interestingly, we observed an increase in hematopoietic multipotent progenitors, with a parallel decrease in erythroid lineage cells (Fig. 7B). These results are consistent with the lineage-specific alterations induced by U2AF1 S34F, with impaired erythroid differentiation[27,76].

By single-cell mutation calling on reads mapping to the U2AF1 locus (see "Methods"), we classified each cell from U2AF1 S34F patient samples as either WT or S34F (Fig. 7C). Notably, we observed that different cell types were characterized by various fractions of mutant cells, ranging from 5% to 32% (Fig. 7D). This data confirm a myelo-monocytic shift with a reduction in megakaryocyte and erythroid

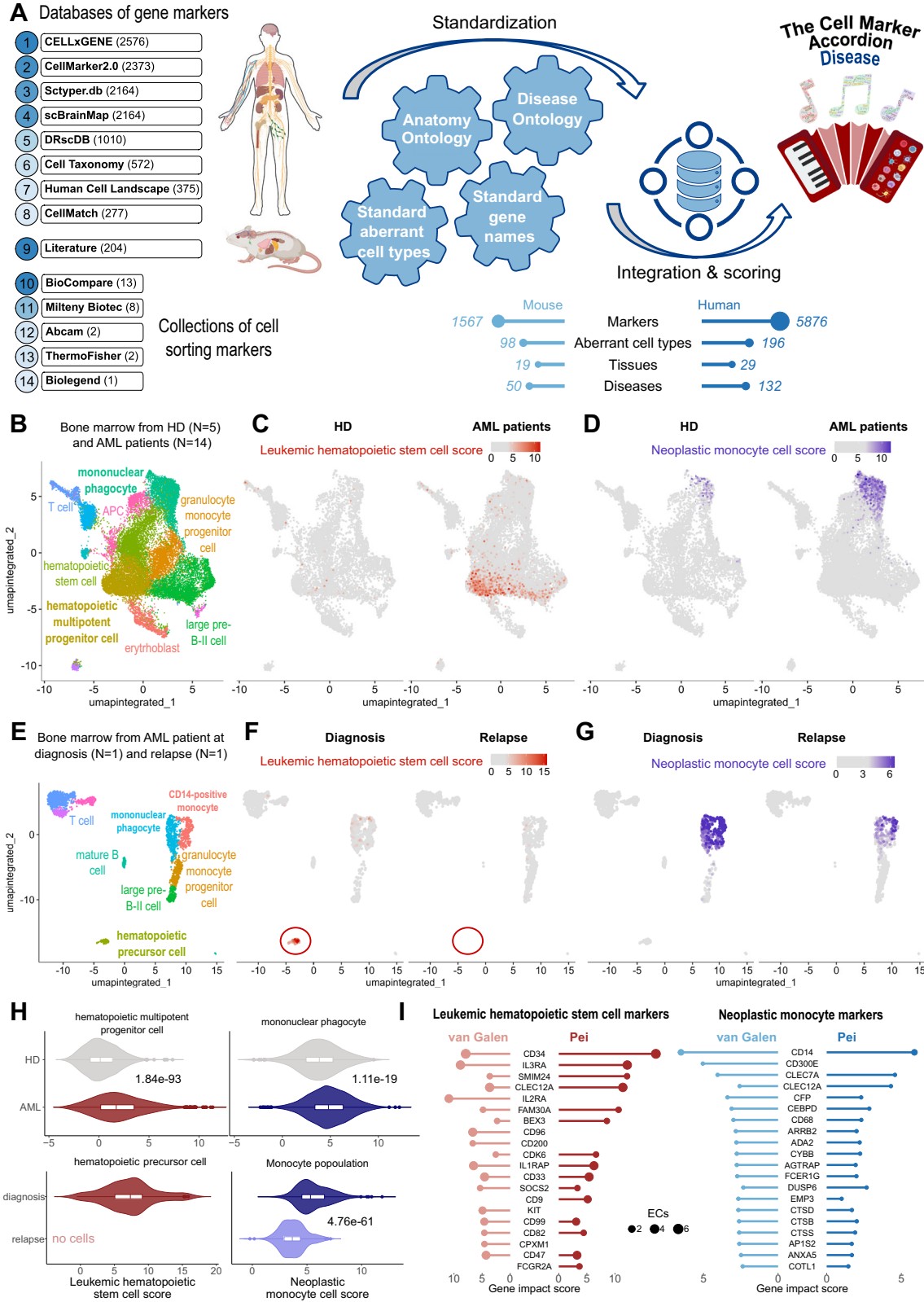

linage priming within hematopoietic stem and progenitor cells from patients with S34F mutant MDS (Fig. 7C); accumulation of mutant cells within the megakaryocytic and erythroid lineage suggests a differentiation defect conferred specifically by the S34F mutation (Fig. 7D).

Taken together, these results demonstrate that the Cell Marker Accordion can effectively identify and dissect cell-type variations driven by pathologically relevant mutations.

## The Cell Marker Accordion identifies the activation of innate immunity pathways in mouse bone marrow

N6-methyladenosine (m6A) is the most abundant eukaryotic internal mRNA modification and significantly influences RNA biology[77–79]. This modification plays an important role in normal hematopoiesis, and alterations in m6A metabolism are strongly associated with acute myeloid leukemia pathogenesis, characterized by the overexpression

**Fig. 5 | The Cell Marker Accordion identifies disease-critical cell types in acute myeloid leukemia patients. A** Workflow for building the Cell Marker Accordion Disease database. The resulting number of human and murine markers for aberrant cell types associated with various diseases from multiple tissues is reported. Mouse and human illustrations created in BioRender. Tebaldi, T. (2025) https://BioRender. com/x09w717. **B** Cell Marker Accordion annotation of human bone marrow cells from healthy donors (HD) and acute myeloid leukemia (AML) patients[59]. **C, D** Identification of leukemic hematopoietic stem cells (LHSCs) (**C**) and neoplastic monocytes (**D**). Cells are colored according to the Cell Marker Accordion scores. **E** Annotation of human bone marrow cells from AML patients at diagnosis and relapse after venetoclax treatment[57]. **F, G** Identification of LHSCs (**F**) and neoplastic monocytes (**G**) in AML patients at diagnosis and relapse. **H** Distribution of LHSC scores in hematopoietic progenitors (left) (HD, $n = 1110$; AML, $n = 3005$; diagnosis,

$n = 89$, relapse, $n = 0$) and neoplastic monocyte scores in monocyte populations (right) (HD, $n = 518$; AML, $n = 2392$; diagnosis, $n = 409$, relapse, $n = 335$) comparing AML patients with healthy donors (top) and AML patients at diagnosis and at time of relapse after venetoclax treatment (bottom). The box plots represent the median as the central line, while the lower and upper hinges correspond to the first and third quartiles (25th and 75th percentiles). Whiskers extend to the smallest and largest values within 1.5 times the interquartile range from the lower and upper quartiles, respectively. One-tailed Wilcoxon Rank Sum test was used, P values are displayed. **I** Comparison of marker genes with the highest impact in defining LHSCs and neoplastic monocytes in the two leukemia datasets, for hematopoietic progenitor cells and monocytes, respectively. Source data are provided as a Source Data file.

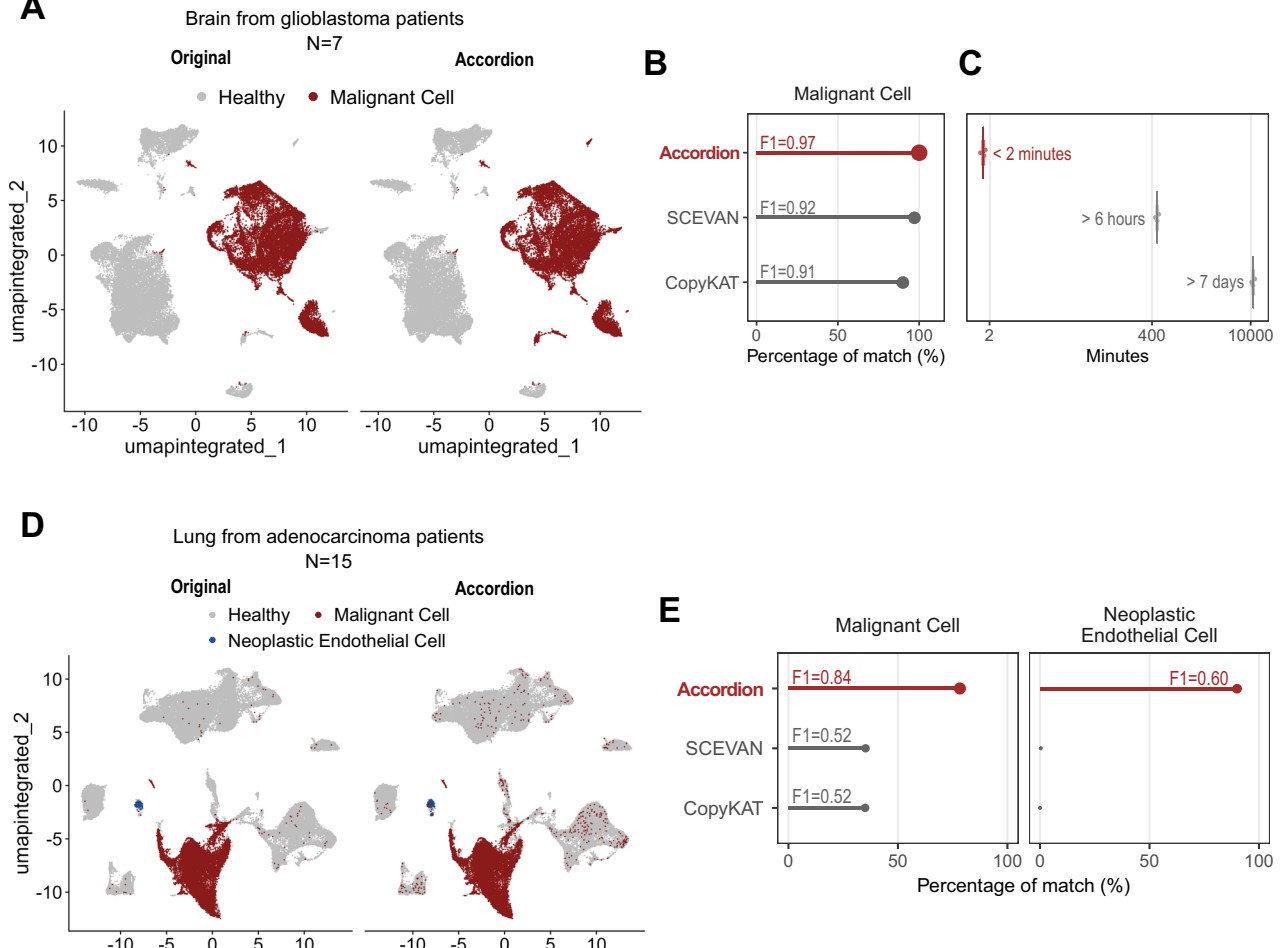

**Fig. 6 | The Cell Marker Accordion improves the identification of malignant cells in solid tumors. A** Identification of malignant cells in glioblastoma patients. Left: original annotation[24]. Right: Cell Marker Accordion annotation. **B** Comparison of annotation performances in identifying glioblastoma malignant cells, measured as the percentage of cells corresponding to the ground truth and the relative F1 scores. Data are presented as mean values ± SEM ($n = 3$). **C** Comparison of

annotation running times among tools. **D** Identification of malignant and neoplastic endothelial cells in lung adenocarcinoma. Left: original annotation[25]. Right: Cell Marker Accordion annotation. **E** Comparison of annotation performances in identifying malignant cells (left panel) and endothelial cells with a neoplastic gene expression signature (right panel). Source data are provided as a Source Data file.

of the m6A methyltransferase METTL3[80,81]. For this reason, pharmacological inhibition of METTL3 has been proposed as a therapeutic strategy to treat leukemias and other tumors[82].

To characterize the effect of m6A modulation on hematopoietic populations, we applied the Cell Marker Accordion to two murine single-cell datasets obtained from the bone marrow of Mettl3 conditional knockout mice[83] (Fig. 8A–D) and from mice upon pharmacological inhibition of METTL3 with STM2457[84] (Fig. 8E–H). After

performing cell type annotation, we compared cell type compositions (Fig. 8B, F). In both datasets, we observed an increase in stem cells and megakaryocyte cells, together with a decrease in erythroid lineages upon Mettl3 deletion or inhibition (Fig. 8C, G). These observations are in line with the original publications and with results obtained in previous studies[28]. Next, we performed cell cycle annotation based on lists of cell-type phase-specific positive and negative markers (Fig. 8B, F, right panels). With this procedure, we could detect cell cycle changes

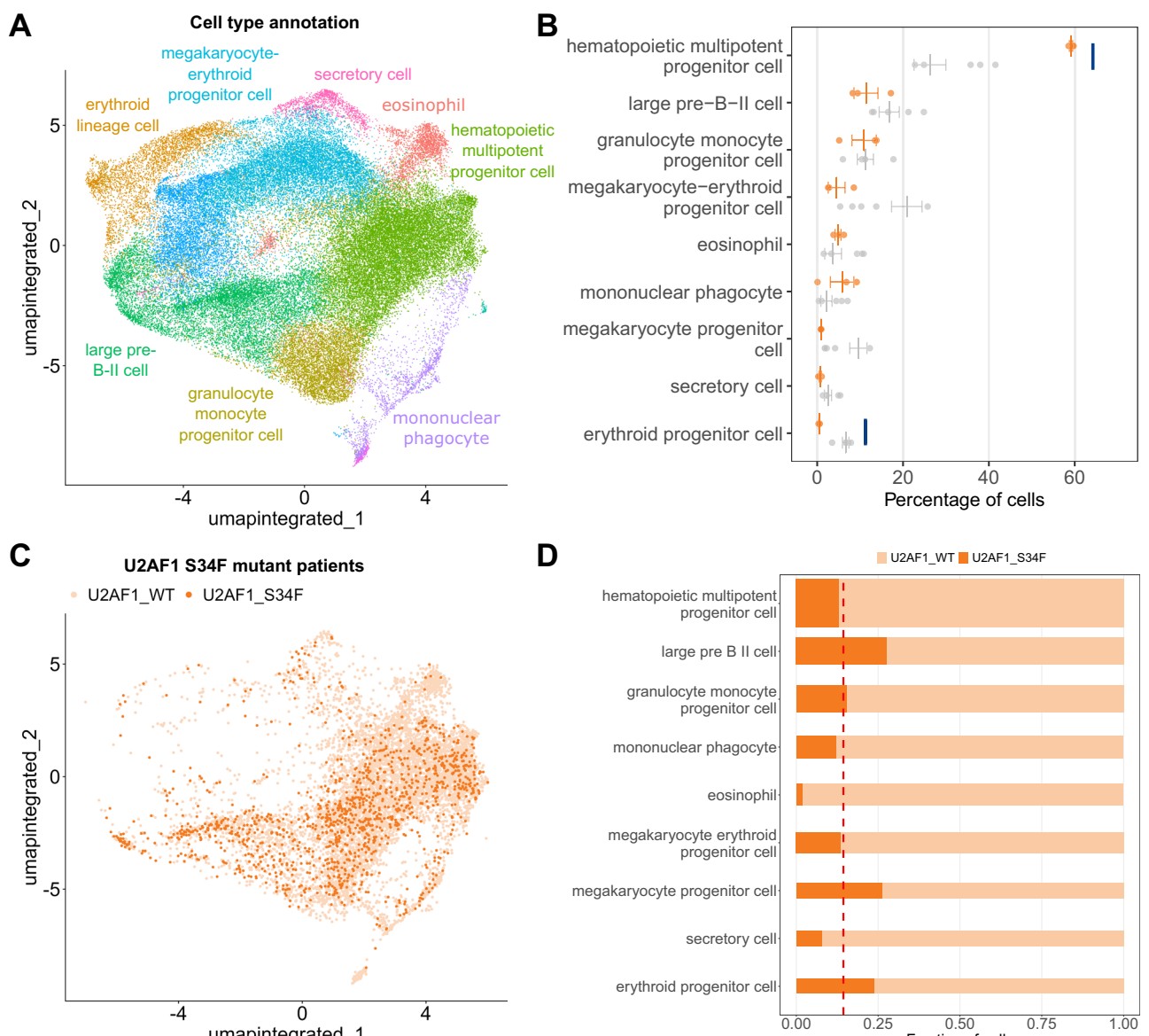

**Fig. 7 | The Cell Marker Accordion identifies cell type alterations in splicing factor mutant cells from patients with myelodysplastic syndromes. A** Cell Marker Accordion cell type annotation of MDS patients with and without U2AF1 S34F mutation. **B** Changes in the abundance of hematopoietic cell types among conditions. Orange bars represent patients with U2AF1 S34F mutations, and gray bars represent patients without splicing factor mutations. Data are presented as mean values ± SEM (U2AF1 WT, *n* = 5, U2AF1 S34F, *n* = 3). Compositional analysis was performed with the scCODA[91] Python package based on Bayesian models. Credible and significant results are highlighted as blue bars, using an FDR threshold of 0.1. **C** Color-code representation of U2AF1 WT and S34F cells in S34F mutant patients. **D** Fraction of mutant (dark orange) and WT cells (light orange) within each cell type. The height of the bar is proportional to the average number of cells in each population. The dashed line represents the average number of mutant cells across all cell types in U2AF1 S34F patients. Source data are provided as a Source Data file.

in specific hematopoietic cell types, in particular, an increase of G0 cells among stem cells and megakaryocytes (Fig. 8D, H).

Two recent studies by us and others turned the spotlight on aberrant activation of innate immune pathways as a consequence of response to the deletion of the m6A methyltransferase Mettl3 or pharmacological inhibition, mediated by the formation of aberrant endogenous double-stranded RNAs[28,29]. To explore the impact of the knockout and the inhibition of Mettl3 on immunity in single-cell datasets, the Cell Marker Accordion computed an "innate immune response" score based on the activation of genes associated with this signature (Supplementary Data 7). Notably, both in the case of Mettl3

KO and drug-induced Mettl3 inhibition, we obtained a significant increase in the innate immune response score with respect to the control condition (Fig. 8I). In addition, by extracting genes that mostly influence the immune response score, we found a subset that exhibits consistent activation in both murine models, as well as sets of genes that are specifically activated in response to either the knockout or the pharmacological inhibition of Mettl3 in murine hematopoietic stem and progenitor cells (Fig. 8J).

Overall, these results demonstrate that the Cell Marker Accordion can be effectively used to characterize pathologically relevant pathways in disease or pharmacological treatment models.

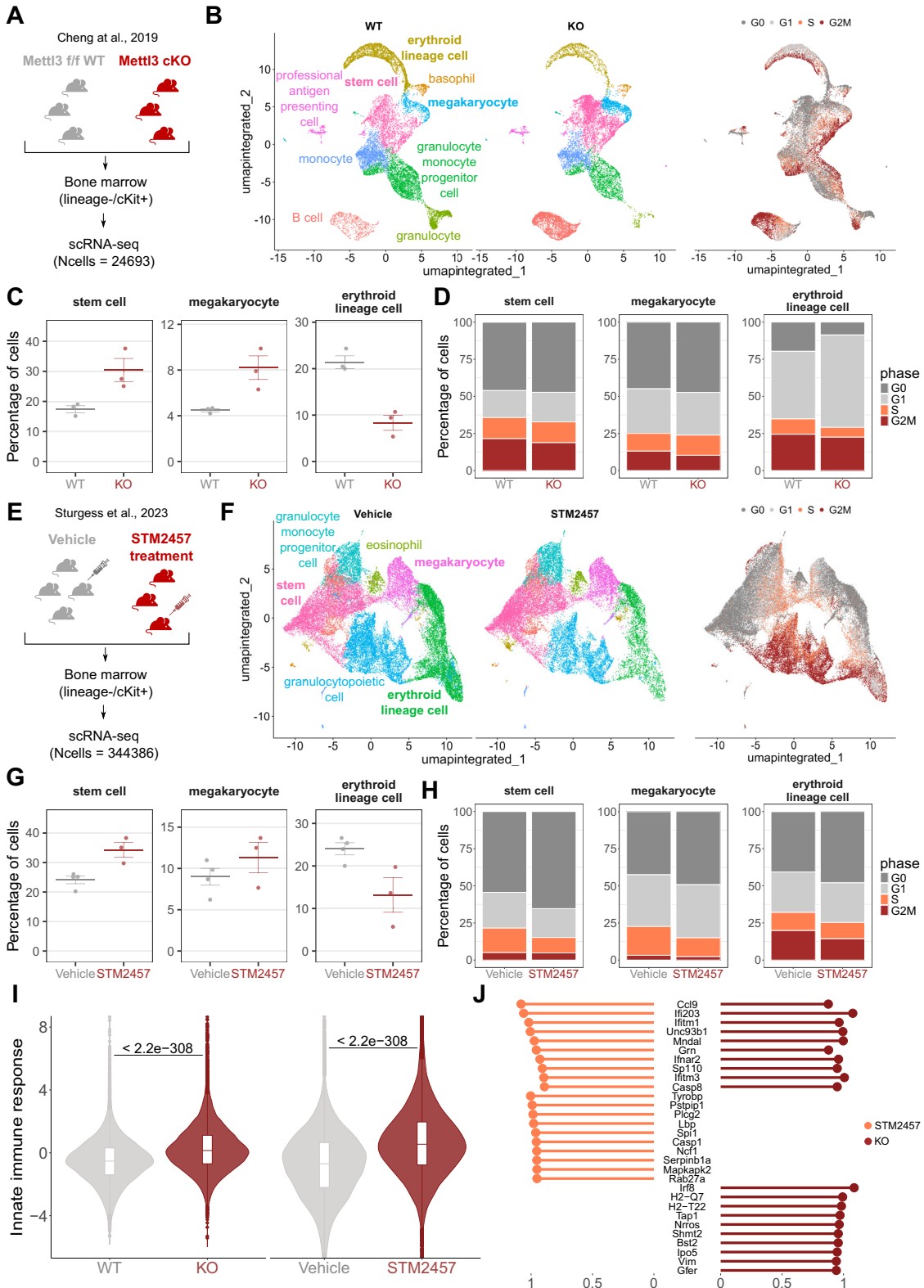

## Discussion

Accurate identification of cell types and states within heterogeneous and complex tissues is a prerequisite for comprehensive exploration and interpretation of single-cell and spatial data to provide biological insights. Yet, it is a challenging step in computational analysis workflows[10].

Here we present the Cell Marker Accordion, a user-friendly platform encompassing an interactive R Shiny web application and an R package designed to automatically identify and easily interpret single-cell populations in both physiological and pathological conditions. With respect to the majority of existing computational methods[12,17,19,40,41,64,65], the Cell Marker Accordion not only provides the

**Fig. 8 | The Cell Marker Accordion identifies activation of innate immunity pathways in mice bone marrow. A** Schematic diagram of the single-cell experimental design of Cheng et al.[83] dataset, comparing bone marrow from Mettl3 KO and WT mice. **B** Accordion cell types annotation of WT and KO mice and identification of cell cycle phase, based on lists of phase-specific markers. **C** Changes in the abundance of specific hematopoietic cell types upon Mettl3 KO. The increase in stem cells and megakaryocytes, with the parallel decrease of erythroid lineages, is consistent with literature. Data are presented as mean values ± SEM (WT, $n = 3$, KO, $n = 3$). **D** Cell type-specific variations in cell cycle between WT and Mettl3 KO bone marrows. **E** Schematic diagram of the Mettl3 inhibition experimental design of Sturgess et al.[84] dataset. **F** Accordion cell types annotation of mice treated with STM2457 METTL3 inhibitor and vehicle-treated mice, and identification of cell cycle phase. **G** Changes in the abundance of specific cell types between STM2457 and vehicle mice, consistent with changes observed in panel (**C**). Data are presented as mean values +/- SEM (Vehicle, $n = 4$, STM2457, $n = 3$). **H** Cell type-specific variations of the cell cycle between STM2457 and vehicle mice. **I** Significant increase of the "innate immune response" signature in Mettl3 KO and STM2457 treated cells, consistent with innate immunity activation observed in Gao et al.[28]. The box plots represent the median as the central line, while the lower and upper hinges correspond to the first and third quartiles (25th and 75th percentiles). Whiskers extend to the smallest and largest values within 1.5 times the interquartile range from the lower and upper quartiles, respectively. One-tailed Wilcoxon Rank Sum test was used, $P$-values are displayed. WT, $n = 12309$; KO, $n = 12015$; Vehicle, $n = 30133$; STM2457, $n = 23443$. **J** Genes involved in "innate immune response" pathways and showing the highest impact score in Mettl3 KO or STM2457 treated cells. Source data are provided as a Source Data file.

user with an accurate annotation of cell types, but it is also able to detect disease-critical cells and pinpoint altered pathways in aberrant conditions, including cell cycle and quiescence analysis. The Cell Marker Accordion combines positive and negative markers, providing a more specific and unambiguous annotation. Cell types can be easily browsed following the Cell Ontology hierarchy and the Uber-anatomy ontology to obtain the desired level of resolution. With respect to existing tools, the Cell Marker Accordion weights markers not only on their specificity but also on their consistency among resources, allowing a more robust cell type identification. Moreover, our tool allows the inclusion of customized annotations, by incorporating any weighted signature of interest. The biological interpretation of results is straightforward and unmatched by existing tools since the Cell Marker Accordion provides detailed information and graphics on genes, cell types or pathways that exert the most significant influence on annotation outcomes.

We validated the performance of the Cell Marker Accordion on 9 single-cell and spatial multi-omics datasets from different tissues, considering surface markers and expert-based annotations as a reference[42–47]. With no exception, the Cell Marker Accordion improved the identification of cell types compared to other available marker-based annotation tools[12,17,19,40,41,64,65], with an average increase of 23% in annotation performance. Additionally, the Cell Marker Accordion is remarkably faster than other tested tools, coupling high annotation accuracy with exceptional computational efficiency.

Accurate identification of cell types is also a fundamental requirement for investigating various pathologies, characterized by disease-critical cells with aberrant gene expression, playing a central role in disease progression and treatment response[20]. Identifying and characterizing disease-critical cells is pivotal for improving diagnosis and interceptive medicine[8], understanding pathogenesis and therapy resistance mechanisms, identifying biomarkers, and developing effective therapies that minimize adverse effects on healthy cells. Current annotation tools focus mostly on physiological cell types. Some approaches, such as SCEVAN and CopyKAT, attempt to distinguish malignant cancer cells in single-cell RNA-seq through CNV calling but cannot be applied to pathologies with a minimal number of genomic alterations, such as leukemias[64,65]. To fill this gap, the Cell Marker Accordion includes weighted collections of marker genes associated with disease-critical cells in multiple common diseases, including solid and liquid tumors. By exploiting scRNA-seq datasets from multiple tumors, we showed that the Cell Marker Accordion effectively identifies aberrant cell types and extracts altered gene signatures, with improved specificity and lower running times than existing tools. While the Cell Marker Accordion's customizability suggests broad applicability in cancer and other diseases, our current benchmark, focused on two liquid and two solid tumors, demonstrates improved performance within these specific contexts. We plan to expand this benchmark in the future as the availability of high-quality, ground-truth datasets increases.

Besides identifying disease-critical cells, the Cell Marker Accordion can be applied to the study and characterization of pathological processes. We demonstrated this in the context of myelodysplastic syndromes (MDS), where mutations in splicing factors (SF) genes such as U2AF1 are prevalent in approximately 50% of patients[66–68] and linked to decreased survival rates[69–75]. By applying the Cell Marker Accordion to single-cell data that we generated from bone marrow of a small cohort of MDS patients, we revealed skewing in the hematopoietic lineages in patients with the U2AF1 S34F mutation. In particular, we observed impaired erythroid differentiation[27,76], pointing out the impact of a pathologically relevant splicing factor mutation on ineffective hematopoiesis and clonal advantage. This approach could be extended by tracking additional MDS mutations to determine their effect at various stages in differentiation.

Finally, we used the Cell Marker Accordion to dissect the effects of m6A RNA modification[77–79] and modulation of the METTL3 methyltransferase on hematopoiesis in murine models. Alterations in m6A have been strongly associated with acute myeloid leukemia pathogenesis[80,81], and pharmacological inhibition of METTL3 has been proposed as a therapeutic strategy[82]. The Cell Marker Accordion identified cell cycle changes and activation of immune response pathways in specific hematopoietic cell types, consistent with the formation of aberrant endogenous dsRNAs upon METTL3 depletion[28,29]. We extracted gene signatures activated in response to either the knockout or drug-mediated inhibition of METTL3 or both to demonstrate that the Cell Marker Accordion can be utilized to characterize pathologically relevant pathways in disease or pharmacological treatment models.

Complementary to the Cell Marker Accordion, several annotation tools exist that do not directly rely on marker genes[85,86]. These methods often rely on correlating reference expression data or transferring learned labels from other annotated single-cell datasets. However, these approaches necessitate high-quality and comprehensive reference datasets with annotated clusters for all relevant cell populations. Importantly, they can be susceptible to technical variations, such as differences in experimental platforms or sequencing strategies[14]. Given the current rate of evolution in single-cell and spatial omics technologies, this aspect cannot be underestimated. For this reason, a direct comparison between these approaches and the Cell Marker Accordion was not conducted in this study.

Fast-forward technological advances are expected to provide increasingly accurate and comprehensive measurements of single-cell and spatial populations. The Cell Marker Accordion is designed to accommodate updates and new sources of information, aiming for a more precise and refined cell type identification in diverse contexts and across different types of data. Possible extensions of the Cell Marker Accordion include single-cell approaches profiling chromatin accessibility[87] or deconvolution of low-resolution spatial omics data.

In conclusion, the Cell Marker Accordion is a user-friendly, fast, flexible and powerful tool that can be exploited to improve the

annotation and interpretation of single-cell and spatial datasets focused on studying diseases.

## Methods

Ethical approval for the study was obtained from the Yale University Human Investigation Committee (protocol number #1401013259). All patients provided written consent in accordance with established guidelines and regulations.

### Data sources of the Cell Marker Accordion database

The Cell Marker Accordion database was constructed by considering multiple published marker gene databases and collections of cell-sorting markers in both physiological and disease conditions (Supplementary Data 1). Literature research was also conducted to collect additional marker genes associated with aberrant cell types for different diseases (each publication is considered a different source in calculating consistency scores). All annotation sources and references are reported in the Cell Marker Accordion database (https://rdds.it/CellMarkerAccordion).

We considered human and mouse marker genes associated with various tissues, including blood, bone marrow, immune system, pancreas, lung, liver, kidney and brain, along with their respective subtypes and associated diseases. Both positive and negative markers, when present, were selected. Marker genes' nomenclature was standardized to ensure the most recent approved version of gene symbols. HUGO Gene Nomenclature Committee (2024) and Mouse Genome Informatics (v. 6.24) resources were employed to standardize human and mouse gene names, respectively. To enable proper integration, we standardized original cell type labels among different sources by mapping them to the Cell Ontology (release 2024-08-16) (http://obofoundry.org/ontology/cl.html)[38], and the NCI Thesaurus OBO Edition (release 2024-05-07) (http://www.ebi.ac.uk/ols4/ontologies/ncit), for physiological and pathology associated cell types, respectively. The Uber-anatomy ontology (Uberon, release 2024-09-03) (https://obofoundry.org/ontology/uberon.html)[39] was used to standardize tissue nomenclature. Finally, the Disease Ontology (release 2024-11-01) (https://disease-ontology.org/)[52] was exploited to standardize disease names.

After standardization and integration, Cell Marker Accordion includes a comprehensive collection of 15479 marker genes associated with 728 cell types in 212 human tissues, and 4118 marker genes associated with 493 cell types in 108 murine tissues (Fig. 2A). Furthermore, the Cell Marker Accordion includes an extensive disease collection of 5876 genes associated with 196 aberrant cell types and 132 diseases in 29 human tissues, and 1567 genes associated with 98 aberrant cell types and 50 diseases in 19 murine tissues (Fig. 5A).

### Overlap among published marker gene databases

To quantify the overlap of marker genes between two marker gene databases (Fig. 1C), we calculated the Jaccard pairwise similarity values, i.e. the ratio between the cardinality of the intersection and the cardinality of the union. This was determined as the average Jaccard similarity of marker genes associated with common cell types present in both databases.

### Definition of integration scores for marker genes in the Cell Marker Accordion database

After integration, the Cell Marker Accordion database describes each marker with a specificity score (SPs) and an evidence consistency score (ECs).

The SPs ranges from 0 to 1 and reflects how many cell types (CT) a gene (G) is a marker for. It is calculated separately for positive and negative markers as:

$$SPs_G = 1/\text{Number of CT with } G \text{ as a marker}$$

A high SPs indicates that marker G is highly specific for a certain cell type, while a low SPs is associated with markers spread among multiple cell types.

The ECs evaluates the agreement of different annotation sources and measures marker robustness and reliability. It is calculated separately for positive and negative markers as:

$$ECs_{G,\,CT} = \text{Number of sources defining gene } G \text{ as a marker of cell type } CT$$

A high ECs indicates a high consensus of marker $G$ among several sources and vice-versa.

### Implementation of the Cell Marker Accordion R package for automatic annotation

The Cell Marker Accordion includes an R package to automatically identify cell type, cell cycle stage and pathway activation in single-cell and spatial omics data (https://github.com/TebaldiLab/cellmarkeraccordion). Users can annotate clusters or cells exploiting the built-in Cell Marker Accordion database with the *accordion()* function or can provide custom sets of markers using the *accordion_custom()* function. Both functions take as input a Seurat[34] object (versions 4 or 5) or a raw count matrix (Supplementary Fig. 1). First, if no prior normalization and scaling steps have been performed, the single-cell expression matrix is normalized and scaled on input marker genes (SE, scaled expression). Based on the input, ECs are computed for each gene G and cell type CT, separately considering positive and negative markers. In case of conflicting sources, i.e. genes listed as positive and negative markers for the same cell type depending on the source, the ECs is penalized: ECs of negative occurrences are subtracted from ECs of positive occurrences to obtain the final ECs, whose sign will determine whether the marker is classified as negative (<0), positive (>0) or not considered (=0), and the final ECs is defined as the absolute value. ECs, ranging from 1 to 23 in the current Accordion database for both positive and negative markers, are log10 transformed to obtain regularized evidence consistency scores (ECs_reg) (Supplementary Fig. 1). Next, the reciprocal of SPs is reverse scaled, separately for positive and negative marker. The result is then scaled to the same range of the ECs and log10 transformed, obtaining the regularized specificity score (SPs_reg). This procedure gives the same overall weight to the two scores. The resulting SPs_reg and ECs_reg are multiplied to obtain a final weight for each marker gene in each cell type. This final weight ranges from 1 to a maximum that depends on the marker database (5.6 in the current Accordion database, a magnitude similar to scaled gene expression levels). The scaled gene expression (SE) level is multiplied by this weight, obtaining a gene-weighted expression score, called gene impact score (G_IMs$_{G,\,C,\,CT}$), for each cell type (CT), each marker gene (G) in CT, in each cell (C):

$$G\_IMs_{G,C,CT} = SE_{G,C} \times ECs\_reg_{G,CT} \times SPs\_reg_G \quad (1)$$

For each cell type, the normalized sum of all associated marker genes is calculated by summing, cell by cell, the weighted expression score divided by the square root of the weighted sum. This step leads to a $CT \times C$ enrichment score matrix, where rows represent cell types, columns represent cells and values indicate the associated cell type impact scores (CT_IMs$_{C,\,CT}$):

$$CT\_IMs_{C,CT} = \frac{\sum_{G \in PG} G\_IMs_{G,C,CT}}{\sqrt{\sum_{G \in PG} ECs\_reg_{G,CT} \times SPs\_reg_G}} - \frac{\sum_{G \in NG} G\_IMs_{G,C,CT}}{\sqrt{\sum_{G \in NG} ECs\_reg_{G,CT} \times SPs\_reg_G}} \quad (2)$$

Where PG and NG are respectively positive and negative markers for the cell type CT. Each cell is then assigned to the cell type with the highest CT_IMs$_{C,\,CT}$. For annotating a cell cluster (CC) the CC cell type

impact score (CT_IMs$_{CC}$) is calculated, by default, as the third quartile of the distribution of the CT_IMs$_{C, CT}$ of cells belonging to CC:

$$CT\_IMs_{CC, CT} = Q3_{C \epsilon CC} \, CT\_IMs_{C, CT} \qquad (3)$$

Each cell cluster is then assigned to the cell type with the highest CT_IMs$_{CC, CT}$, i.e the winning cell type (Supplementary Fig. 1). When annotating cell clusters, the CC gene impact score (G_IMs$_{CC}$) is calculated by default as the third quartile of the distribution across all cells belonging to the same cluster. Additionally, at both cell and cluster annotation resolution, the cell type gene impact score is calculated, by default, as the third quartile of the G_IMs$_{G, C, CT}$ distribution across all cells belonging to the same winning cell type.

### The Cell Marker Accordion Shiny app

The R Shiny tool is available at https://rdds.it/CellMarkerAccordion/. The Shiny app incorporates reactive programming, allowing users to access the Cell Marker Accordion database and to retrieve marker genes that are specific for their selected cell types and tissues. Furthermore, when users choose genes of interest on the marker gene tab, the tool interactively retrieves the standardized cell types associated with the selected genes.

### Validation of the Cell Marker Accordion

To validate the Cell Marker Accordion, we exploited eight single-cell and multi-omics datasets from different healthy tissues along with a spatial transcriptomic brain dataset (Supplementary Data 4). For each dataset, the original cell type annotation was considered as ground truth. Cell type nomenclature was mapped to the Cell Ontology if necessary (Supplementary Data 4). After the preprocessing steps, each cluster was assigned to the cell type corresponding to the largest fraction of cells within the cluster. This approach ensures that the assignment reflects the predominant cell type composition within each cluster. Annotation performance was compared against five automatic annotation tools, ScType[40], SCINA[19], clustifyR[12], scCATCH[41], scSorter[17]. The Cell Marker Accordion was run using max_n_marker set to 30 and specifying the tissue of the dataset. All other tools were run using their default parameters (Supplementary Data 5). scType and scCATCH have in-built gene marker databases, which were used to run the tools. SCINA, clustifyR and scSorter require a user-provided set of marker genes in input: positive markers from the Cell Marker Accordion database were used in the benchmark. To avoid running failures, a maximum of 100 marker genes for each cell type was set for SCINA. For each tool, the annotation performance is measured as the average percentage similarity between the predicted cell type and the ground truth for each cluster. Percentage similarity between cell types is determined using the Cell Ontology tree and ranges from 0 to 100, where 0 represents the maximum possible distance, and 100 indicates a perfect match. This similarity was computed using the get_sim_grid function from the ontologySimilarity R package (version 2.7). Running time was also tested on the Zheng et al.[42] dataset. To provide an unbiased comparison, we considered the same list of marker genes as input for all methods and ran the tools on the same machine three times. The benchmark was conducted on a server with: CPU: 2x Intel(R) Xeon(R) Gold 5318Y CPU @ 2.10 GHz; RAM: 1536 GB DDR4. The tests were run on Ubuntu 22.10 with R v4.4.2. All tools were configured to use 4 threads. Running for over 3 h without completion, scSorter was labeled with "Timeout" and excluded from further benchmarks.

To validate the Cell Marker Accordion on the annotation of disease-critical cells, we took advantage of two published single-cell datasets[24,25] of glioblastoma and lung adenocarcinoma patients samples. In the glioblastoma dataset, tumor cell clusters were identified based on the expression of a gene list provided by the original authors, which included markers such as SOX2, OLIG1, GFAP, and S100B. We identified clusters 1, 4, 5, 7, 10, 11, 12, 20, 21 as tumor cells

(Supplementary Fig. 7) and used them as ground truth. The lung adenocarcinoma dataset was already annotated with malignant cell types, which were used as ground truth. Annotation performances were calculated as the percentages of correctly identified aberrant cells and the corresponding F1 scores, based on precision and recall. The Cell Marker Accordion performances and running times were compared against SCEVAN[64] and CopyKAT[65], run with default parameters (Supplementary Data 5).

### MDS single-cell dataset

Human primary cells were obtained with patients' written consent after approval by the Yale University Human Investigation Committee. Bone marrow samples from MDS patients (Supplementary Data 6) were processed as also reported in Biancon et al.[88]. Primary mononuclear cells from MDS patients were thawed in a 37 °C water bath. Once thawed, cells were spun down at 500 × g for 5 min and resuspended in 10 ml of FACS buffer (PBS,0.5% BSA, 2 mM EDTA). Cells were counted and spun down again at 300 × g for 10 min. Cells were stained with Pacific Blue anti-human CD34 (BioLegend, Cat# 343512; 1:100) for CD34$^+$ blast isolation and 7-AAD (STEMCELL Technologies, Cat# 75001; 1:100) for viability evaluation in 500 µl FACS buffer and incubated at 4 °C for 30 min. One million unstained cells were reserved as controls for each patient sample. Next, cells were washed with 10 ml FACS, spun down at 300 × g for 10 min at 4 °C, and resuspended in 200 µl FACS buffer per million cells. The cell suspension was pipetted through a 70 µm filter into a 5 ml tube for sorting by the Yale Flow Cytometry facility on the FACSAria instrument (BD Biosciences). Specifically, a gating strategy to isolate viable (7−AADneg) MDS blasts (CD34pos) was applied. Sorted cells were subsequently processed for scRNA-seq library preparation by the Yale Center for Genome Analysis using Chromium Next GEM Single Cell 5' kit v2 (10x Genomics).

Single-cell expression counts were derived from raw FASTQ files using the 10x Genomics Cell Ranger pipeline v7.1.0 with default parameters and the GRCh38 (2020-A) reference genome. A total of 64,915 cells with an average of 71,828 reads and 3476 genes were obtained. Variant detection at position 21:43104346-43104346 (U2AF1 S34F) was performed using FreeBayes v1.2.0 (https://github.com/freebayes/freebayes). Variant annotation was performed using VarTrix v1.1.19 (https://github.com/10XGenomics/vartrix). To ensure the removal of empty droplets and potential doublets, cells with less than 200 expressed genes and more than 6000 were filtered out. Moreover, cells with 20% mitochondrial-derived counts were removed, resulting in a total of 62,496 high-quality cells for downstream analysis. Data were first log-normalized, and the most 2000 highly variable features were identified. Scaling was then performed on all genes. Linear dimensionality reduction through Principal Component Analysis (PCA) was performed on scaled data and scRNA-seq cells were clustered based on a shared nearest neighbor graph (obtained with the FindNeighbors function) employing the FindClusters function with the default Louvain algorithm for community detection, with 0.5 resolution. For cluster visualization, Uniform Manifold Approximation and Projection for Dimensionality Reduction (UMAP) was employed. The IntegrateLayers function was used to align cells across multiple samples by correcting batch effects while preserving biological variability, using the RPCA integration method (k.weight = 70). Clustering was performed on the integrated data, and cells with similar gene expression profiles were grouped into distinct clusters. A resolution of 0.4 was set, resulting in 13 clusters. Cell type annotation was performed using the Cell Marker Accordion, with bone marrow selected as tissue. Parameters were set to max_n_marker = 30, log2FC_threshold = 1, and allow_unknown = FALSE.

### Analysis of published single-cell data

Single-cell datasets were analyzed employing a standard preprocessing pipeline for single-cell RNA-seq data, using Seurat (version 5.0.3) in the R environment (version 4.2.3). Data were first log-

normalized (for ADT data, centered log ratio (CLR) was applied) and the most highly variable features were identified and scaled. Linear dimensionality reduction through Principal Component Analysis (PCA), was performed on scaled data and scRNA-seq cells were clustered based on a shared nearest neighbor graph (obtained with the FindNeighbors function) employing the FindClusters function with the default Louvain algorithm for community detection. For cluster visualization Uniform Manifold Approximation and Projection for Dimensionality Reduction (UMAP) was employed. In case of scRNA-seq datasets integration, the IntegrateLayers function was used to align cells across multiple samples by correcting batch effects while preserving biological variability, using the RPCA integration method. Specific parameters for the analysis of each dataset can be found in the Supplementary Data 4.

### Analysis of published spatial data

The MERFISH mouse brain dataset was analyzed employing a standard pre-processing pipeline for imaging-based spatial dataset, using Seurat (version 5.0.3) in the R environment (version 4.2.3). Data were first normalized with SCT-transform. Linear dimensionality reduction through Principal Component Analysis (PCA), was performed on transformed data and cells were clustered based on a shared nearest neighbor graph (obtained with the FindNeighbors function) employing the FindClusters function with the default Louvain algorithm for community detection. For cluster visualization, Uniform Manifold Approximation and Projection for Dimensionality Reduction (UMAP) was employed. The mouse coronal brain section with ID "Zhuang-ABCA-1.080" was selected based on the highest number of barcodes (37,068). Spatial mapping of cells was performed using NIfTI images, aligning the spatial coordinates of the cells to the Allen CCF 2020 with a voxel spacing of 0.01 mm. Specific parameters for this analysis can be found in the Supplementary Data 4.

### Statistics and reproducibility

This study utilized data from previously published sources, with details on study design available in the original publications referenced in Supplementary Data 4. The statistical analyses conducted are outlined in the "Methods" section and in figure captions, and the results are reproducible by following the described procedures. For the MDS dataset, we included in our cohort eight patients, five patients without splicing factor mutations, and three patients with the U2AF1 S34F mutation. No data exclusions. To assure reproducibility, at least three patient samples per condition were analyzed. Blinding is not relevant in this study based on single-cell transcriptome profiling of U2AF1-mut vs wildtype patients.

### Reporting summary

Further information on research design is available in the Nature Portfolio Reporting Summary linked to this article.

## Data availability

The Cell Marker Accordion gene marker database is available as an Excel file (Supplementary Data 8). Additionally, it is available in the GitHub repository of the Cell Marker Accordion Shiny web app (https://github.com/TebaldiLab/shiny_cellmarkeraccordion). The curated Cell Marker Accordion database can also be downloaded from the online Shiny web app (https://rdds.it/CellMarkerAccordion/) by clicking the "Download" button in the sidebar. The GitHub repositories for both the Shiny web app and the R package (https://github.com/TebaldiLab/cellmarkeraccordion) contain instructions on accessing and downloading the Cell Marker Accordion database.

The Cell Marker Accordion database can be easily explored or customized using the Shiny web app (https://rdds.it/CellMarkerAccordion/), which allows to: a) search and download lists of marker genes associated with input cell types across different tissues in health and disease; b) search and download lists of cell types associated with input marker genes across different tissues in health and disease; c) integrate custom set of marker genes with the Cell Marker Accordion database; d) perform cell type marker enrichment analysis across tissues in health and disease. Additionally, users can also browse hierarchies of cell types following the Cell Ontology structure in order to obtain the desired level of specificity in the markers and rank and select marker genes by their evidence consistency and specificity scores.

Demo examples are provided for guidance. All resulting tables can be easily explored and downloaded. A tutorial to the web app's functionalities, along with instructions for local installation, is available on the GitHub page of the Shiny app: https://github.com/TebaldiLab/shiny_cellmarkeraccordion. Count matrices of patient samples generated by CellRanger v7.1.0, along with U2AF1 S34F mutation information from VarTrix v1.1.19, are available in Zenodo (https://doi.org/10.5281/zenodo.15241428). Due to privacy regulations, raw sequencing files from patients are available upon request. All publicly available scRNA-seq and spatial datasets used in this study are listed in Supplementary Data 4, and the corresponding Seurat objects are deposited in Zenodo (https://doi.org/10.5281/zenodo.15241428). Source data are provided as a Source Data file.

## Code availability

The Cell Marker Accordion R package, with code, tutorial and documentation, is available at https://github.com/TebaldiLab/cellmarkeraccordion under MIT license, and in Zenodo (https://doi.org/10.5281/zenodo.15403792)[89]. To run the Cell Marker Accordion Shiny app locally, the code is available at https://github.com/TebaldiLab/shiny_cellmarkeraccordion under MIT license and in Zenodo (https://doi.org/10.5281/zenodo.15412280)[90].

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

## Acknowledgments

We thank Guilin Wang, Christopher Castaldi and the Yale Center for Genome Analysis for scRNA-seq guidance. We thank Lesley Devine and the Yale Flow Cytometry Facility for guidance in cell sorting. We thank all our patients and all clinical staff for their help with patient recruitment. We thank Emanuele Filiberto Rosatti and Roan Spadazzi for their valu-able comments. This study was funded by AIRC under MFAG 2020 (ID. 24883 project) to T.T. E.B., and T.T. were also supported by Fondazione VRT ("bando intelligenza artificiale 2024"). T.T. has been supported by the MUR PNRR project CN RNA> RINGTAIL (CN00000041), M4C2 Inv 1.4, funded by the NextGenerationEU. S.H. was supported by NIH/NIDDK R01DK124788, U54DK106857, NIH/NCI R01CA266604, NIH/NCI R01CA222518, NIH/NCI R01CA253981, U01CA294514, The Frederick A. Deluca Foundation and the Edward P. Evans Foundation. G.B. was sup-ported by AIRC under Start-Up 2023 - ID. 29035 project, the American Society of Hematology Scholar Award, and the Edward P. Evans Foun-dation. G.V. and F.L. were supported by the European Union within the MUR PNRR 'National Center for Gene Therapy and Drugs based on RNA Technology' (Project no. CN00000041 CN3 RNA). M.C.M. and V.B. were supported by AIRC IG 2021 no. 25704. This work was also supported by the "Departments of Excellence 2023-2027" initiative (Law 232/2016), project no. 40613, funded by the Italian Ministry of University and Research (MUR). We are also grateful to AIL Trento and AIL Bolzano for their generous financial support.

## Author contributions

Conceptualization: E.B., G.Bi., S.H., and T.T.; methodology: E.B., G.Bi., F.L., Z.I., G.V., S.H., and T.T.; investigation: E.B., G.Bi., I.C., K.R.A, J.V., T.S., S.H. and T.T.; formal analysis: E.B., G.Bi., I.C., F.L., Z.I., C.R., G.T., S.H., and T.T.; resources: S.H. and T.T.; validation: E.B., G.Bi., I.C., K.R.A, J.V, T.S., S.H., and T.T; writing: E.B., G.Bi., S.H., and T.T.; visualization: E.B., G.Bi., I.C., S.H., and T.T.; supervision: S.H. and T.T.; project administration: S.H.

and T.T.; funding acquisition: S.H. and T.T.; revision: E.B., G.Bi., G.T., C.R., M.C., G.Bu., A.P., S.M.M., V.B., F.R., L.T., M.C.M., P.M., S.H., and T.T.

## Competing interests

S.H., consultancy, Forma Therapeutics. Other authors declare no competing interests.

## Additional information

[1]Laboratory of RNA and Disease Data Science, Department of Cellular, Computational and Integrative Biology (CIBIO), University of Trento, Trento, Italy. [2]Section of Hematology, Department of Internal Medicine, Yale Comprehensive Cancer Center, Yale University School of Medicine, New Haven, CT, USA. [3]Hematology Unit, Fondazione IRCCS Ca' Granda Ospedale Maggiore Policlinico, Milan, Italy. [4]Institute of Biophysics, CNR Unit at Trento, Trento, Italy. [5]Laboratory of Experimental Cancer Biology, Department of Cellular, Computational and Integrative Biology (CIBIO), University of Trento, Trento, Italy. [6]Armenise-Harvard Laboratory of Brain Disorders and Cancer, Department of Cellular, Computational and Integrative Biology (CIBIO), University of Trento, Trento, Italy. [7]Department of Clinical and Molecular Medicine, Norwegian University of Science and Technology (NTNU), Trondheim, Norway. [8]Laboratory of Molecular and Cellular Neurobiology, Department of Cellular, Computational and Integrative Biology (CIBIO), University of Trento, Trento, Italy. [9]Department of Pathology, Yale University School of Medicine, New Haven, CT, USA. [10]These authors contributed equally: Emma Busarello, Giulia Biancon. ✉e-mail: emma.busarello@unitn.it; stephanie.halene@yale.edu; toma.tebaldi@unitn.it

