## [Transparent Peer Review file · Nature Communications]

Cell Marker Accordion: interpretable single-cell and spatial omics annotation in health and disease

Corresponding Author: Professor Toma Tebaldi

Version 0:

Reviewer comments:

Reviewer #1

(Remarks to the Author)

The manuscript describes the use of the Cell Marker Accordion for annotating and interpreting single-cell populations in various tissues, with a primary focus on hematopoietic tissues in both physiological and pathological states. While the paper presents interest for the community, the manuscript exhibits several significant limitations.

1. The primary limitation is the tool's focus predominantly on hematopoietic tissues. This restricts its applicability across a broader range of tissue types like the brain, lung, pancreas, or kidney, which are critical in many disease studies and fundamental biological research. Expanding its utility to include these and other tissues would make the tool more versatile and valuable across different fields of biomedical research. Consequently, it remains unclear how the Cell Marker Accordion surpasses ScType (<https://www.nature.com/articles/s41467-022-28803-w>), which supports annotations for more than 20 tissue types, automatically assigns cell types, and integrates multiple datasets and public sources.

2. While Cell Marker Accordion allows for identification of disease-critical single-cell populations in several cancer types, it is not clear how it surpasses tools like SCEVAN (<https://www.nature.com/articles/s41467-023-36790-9>), which supports more tissue types and is robustly validated in recent studies. A comparative analysis highlighting specific cases where the Cell Marker Accordion provides superior results in terms of accuracy, user experience, or unique features would greatly enhance its positioning within the research community.

3. The manuscript lacks a comprehensive comparison with other state-of-the-art single-cell annotation tools. Such comparative analysis, including strengths, weaknesses, and performance benchmarks, would provide a clearer positioning of the tool in the landscape of single-cell analysis technologies. Detailing scenarios where it outperforms or falls short against other tools would help potential users make informed decisions.

Minor points:

1. Although Figure 1 illustrates the variability in cell type annotations between CellMarker and PanglaoDB, ScTypeDB appears to effectively integrate and curate these databases, resulting in annotations that closely match those reported in the original article.

2. While the Cell Marker Accordion is marketed as an R package, it requires both a Python installation and a properly configured virtual environment for successful installation on Windows. This requirement should be explicitly stated and accompanied by comprehensive, step-by-step installation instructions, as the lack of such details currently makes the package non-functional on Windows systems without these prerequisites (based on tests conducted on three different machines).

(Remarks on code availability)

While the Cell Marker Accordion is marketed as an R package, it requires both a Python installation and a properly configured virtual environment for successful installation on Windows. This requirement should be explicitly stated and

accompanied by comprehensive, step-by-step installation instructions, as the lack of such details currently makes the package non-functional on Windows systems without these prerequisites (based on tests conducted on three different machines).

Otherwise step-by-step guide was clear and easy to use.

Reviewer #2

(Remarks to the Author)

In this manuscript, the authors introduce the Cell Marker Accordion, a website and R package designed to improve the annotation and interpretation of scRNA-seq data. Traditional tools and databases often yield inconsistent cell type annotations due to variability in marker genes, particularly in pathological contexts. The Cell Marker Accordion addresses this important issue by integrating multiple gene marker databases and literatures, and weighting markers by specificity and evidence consistency. Validation across multiple hematopoietic cell lineage datasets demonstrated improved accuracy in cell type identification, including the ability to detect disease-critical cells in conditions like acute myeloid leukemia and multiple myeloma. This method holds the potential to become a powerful tool for studying hematopoietic cell lineage scRNA-seq datasets under both physiological and pathological conditions. However, there are some concerns in Cell Marker Accordion, particularly for its scope of application in diverse tissues and the lack of benchmarking with published methods. The comments are below.

Major:

1. The scope of application in diverse tissues is a serious concern for Cell Marker Accordion. Although the authors demonstrated the application of Cell Marker Accordion in several datasets in the manuscript, all the included datasets were generated from hematopoietic cell lineages. The performance of Cell Marker Accordion in scRNA-seq datasets generated from all other tissues/organs remains unknown. If the authors aimed to introduce Cell Marker Accordion as a general-purpose method for cell type annotation, its performance in diverse tissues should be evaluated. Otherwise, the authors at least need to clearly state in the abstract and discussion that Cell Marker Accordion is only tested on and applicable to hematopoietic cell lineages.
2. Additional benchmarking with published single-cell annotation methods is necessary to evaluate the performance of Cell Marker Accordion. There are many published methods for automatic annotating cell types in scRNA-seq data. However, this study only benchmarked Cell Marker Accordion with several cell marker gene datasets, but not benchmarked with bioinformatic methods for automatic annotating cell types. Please consider benchmarking Cell Marker Accordion with cell type annotation methods mentioned in two benchmarking studies: PMID: 31500660 and 33359678.
3. The title "Interpreting single-cell messages in health and disease with the Cell Marker Accordion" does not summarize the work well. The meaning of "Interpreting single-cell messages in health and disease" is not clear to readers. The authors may change the title that appropriately reflect the functions and/or findings of Cell Marker Accordion.
4. The calculation of specificity and EC score should take positive and negative markers into the consideration. In the Methods section "Definition of integration scores for marker genes in the Cell Marker Accordion database", there is no description of how to handle positive and negative markers. In particular, are there some genes listed as positive markers in one source and listed as negative markers in the other source? These genes should be considered conflicting evidence but not consistent evidence when calculating specificity and EC score.

Minor:

1. In the Results section "Widespread heterogeneity across annotation sources leads to inconsistent cell type annotation", it is not clear how Jaccard similarity among marker gene databases are calculated. It is suggested to add some paragraphs in Methods to describe what cell types and what marker genes were used to calculate Jaccard similarity, and please provide the marker genes from datasets as a supplementary file if possible.
2. Figure 4A, it is not clear what marker genes and literatures for AML were used by Cell Marker Accordion. Please cite the relevant literatures and report the markers in a supplementary file and/or on shiny App.
3. Supplementary Figure 5: it is better to add a UMAP plot showing the cell cluster labels, which are shown in Supplementary Figure 5E but not in UMAP.
4. Sentence "By single-cell mutation calling on reads mapping to the U2AF1 locus (see Methods)", there is no methods for mutation calling in the manuscript.
5. Figure 5B, it is better to show the cell percentage in individual patients from two groups (5 from SF_WT and 3 from U2AF1), because the merged percentage may be misleading when there is strong intra-group variation. Figure 5C-D, using two contrasting colors would be better to distinguish the two groups.
6. Similar to the comment 5 above, Figure 6C,D,G,H, it is better to show the cell percentage in individual mice from the two groups in each study.
7. Please add a Data Availability statement for the data generated in this study.

(Remarks on code availability)

Please find the attached code review report to improve the R package.

Reviewer #3

(Remarks to the Author)

Gene marker-based methods are commonly used to annotate cell types in scRNA-seq data. However, depending on the list of gene markers assigned to each cell type and the gene marker databases used, the results can be highly variable even if

the same annotation algorithm is applied. In this study, Busarello et al. have addressed this important problem by developing a novel method called Accordion, which takes advantage of using multiple gene marker databases to improve cell type annotations results. This represents an important step forward in the field.

Major Comments:

Comment 1: This study has focused on immune cells and hematopoietic cell types. It is well known that inconsistencies in cell type annotation results due to gene markers are a general problem that applies to many different tissue types, not just immune cells. While single-cell RNA-seq data and relevant gene markers are available for a wide range of cell types and tissues, my question is: "why hasn't Accordion been developed for a broader range of cell types and tissues? What factors might have limited the implementation of Accordion for other cell types, and its utility for broader and more general applications? Are there any plans to extend this study to non immune related cell types and diseases?"

Comment 2: When considering the improvement in matching with FACS populations, the results have substantially improved for PBMCs from Zheng et al. 2017 with improvement in percentage of match > 20% (Figure 3B); however, the improvements are more subtle for the bone marrow data (Triana 2021) and the multi-study scRNA-seq data (comprising a census of immune cells, bone marrow, and umbilical cord blood datasets) (Improvement in percentage of match is around 5-8%) (Supplementary Figures 3A and B). It is important to discuss where and under what conditions this approach would work best and realize its full potential, and what factors are necessary to achieve substantial improvements in the results.

Comment 3: I appreciate that the authors have tested Accordion across several datasets and demonstrated its utility with a wide range of applications (healthy individuals, disease-relevant cells; human and mouse data). However, for the case studies presented in Figures 4-6, it is important to see how other single annotation methods perform and how much improvement (in terms of biological discoveries) is gained by using Accordion compared to single databases. Are the other single database methods capable of making similar discoveries, or do they fail? This comparison will help us understand the extent of the improvement gained from Accordion compared to single-database approaches. Is it a subtle enhancement, or is it substantial?

Comment 4: The incorrect or inconsistent cell type annotation issue presented in Figure 1 can be influenced by the chosen cell type annotation algorithm (in this case, scType). It would be interesting to test this across multiple annotation algorithms (e.g., SCINA, clustifyR, scCATCH, scSorter) and present the level of similarity (measured by Jaccard) across single databases for various annotation algorithms, not just scType.

Minor Comments:

Page 9, in the sentence "which are one of the the key factors contributing". "The" is repeated. Please correct it.

On Page 19, for the following sentence: "Then, both specificity and EC score are log transformed and multiplied to obtain a comprehensive weight for each marker gene in each cell type."

1 - Could you provide a more complete justification for measuring the weight in this way? What is the range of values for the calculated weights?

2 - Is "i." after the above sentence a typo?

In the last paragraph of Page 3, it is mentioned: "... with an average Jaccard similarity of 0.04 and a maximum of 0.12." However, in Figure 1C, it appears that the Jaccard similarity is around 0.5. These two values seem inconsistent.

I suggest that the formulas presented in Supplementary Figure 1 be included in the Methods section (first half; Page 18), so that the reader can gain a more complete understanding of the details related to the annotation and scoring workflow.

(Remarks on code availability)

The README file on the GitHub page provides sufficient detail to run the code. I was not successful in installing cellmarkeraccordion on my machine due to discrepancies with the dependencies, but that may be an issue on my end rather than a problem with the package.

Version 1:

Reviewer comments:

Reviewer #1

(Remarks to the Author)

The authors made a thorough revision of the manuscript. I am generally pleased with the improvements, particularly regarding the installation issues and usability of the GitHub repository. The Cell Marker Accordion now demonstrates broader applicability across tissues and improved benchmarking. However, I have several remaining concerns:

1. While the integrated and curated database represents a major contribution to the field, I didn't find a simple way to download it. The text mentions the database is available through the Shiny app, R package data folder, and GitHub, but none of these provide straightforward access to the raw data. Given that many researchers may want to integrate these valuable marker genes with their own analyses or computational pipelines without needing to use the full R package, I strongly recommend providing direct downloads of the curated database as simple Excel/CSV files. These should be

separated into normal human tissues, normal mouse tissues, human disease states, and mouse disease states. Such accessibility would significantly enhance the utility of this important resource for the scientific community.

2. Disease State Generalizability: While the authors expanded the disease database significantly, I notice that the validation focuses heavily on certain cancer types. To claim broad applicability across malignancies, the authors should either: a) Demonstrate validation across more diverse cancer types and tissues, or b) More carefully scope the claims about malignancy detection to the specific cancer types they have validated

Minor Points:

1. Consider adding a brief "Data Access Guide" section to the manuscript describing the different ways researchers can access and utilize the curated database.
2. In the GitHub repository, include example scripts showing how users can integrate their own marker genes with the database.
3. For transparency, please include in the supplementary materials a complete list of all sources used for database curation, including how each source was processed and integrated.

(Remarks on code availability)

The code was easy to use, but I didn't find an easy way to access the database.

Reviewer #2

(Remarks to the Author)

During the revision the authors made a significant improvement on Cell Marker Accordion to extend its scope from hematopoietic cell lineage to diverse tissues. In addition, Cell Marker Accordion was benchmarked with several single-cell annotation methods and demonstrated its strength. However, there are still some minor issues:

1. The authors mapped cell types to Cell Ontology terms and tissues to Uberon terms. This is a good standardization step, but no details were provided. A table of the mapping relationship may be useful to show the relevant information. Please consider adding this information to the webserver (better for webserver users) or as a supplementary table.

2. The webserver requires improvements and bug fixes in the following areas:

Search by Tissue and Cell Types – When using this feature, I randomly selected various tissues and cell types. However, after repeating this process approximately five times, I was disconnected from the server.

Cell Type Annotation – I uploaded the required differentially expressed gene table, but encountered the error message during annotation: "An error has occurred. Check your logs or contact the app author for clarification." This issue persists regardless of the parameters set.

(Remarks on code availability)

Version 2:

Reviewer comments:

Reviewer #1

(Remarks to the Author)

(Remarks on code availability)

Reviewer #2

(Remarks to the Author)

The authors have fully addressed the previous comments. The bugs are fixed. So I suggest acceptance.

(Remarks on code availability)

The codes are working by this review.

We are pleased to submit a revised manuscript, in response to all the reviewers' comments.

To address the reviewers' valuable feedback, we have:

1. **Expanded the tool's applicability:** we significantly broadened the scope of the Cell Marker Accordion, previously limited to hematopoietic tissues, to encompass hundreds of human and murine tissues.
2. **Enhanced benchmarking:** we conducted comprehensive comparisons with other leading annotation tools across diverse single-cell and spatial omics datasets, including both physiological and pathological contexts, and confirmed the improved performance of the Cell Marker Accordion.

These enhancements have resulted in substantial revisions to the original manuscript: all the main and supplementary figures were updated to reflect the changes in the Cell Marker Accordion. We included two entirely new main figures to showcase the improved benchmarking results, now based on 11 different datasets. We also provide data source files for our figures. Major changes to the text are highlighted in the revised manuscript.

We believe these revisions have significantly strengthened the manuscript, and we sincerely appreciate the reviewers' insightful comments.

REVIEWER COMMENTS

Reviewer #1 (Remarks to the Author):

The manuscript describes the use of the Cell Marker Accordion for annotating and interpreting single-cell populations in various tissues, with a primary focus on hematopoietic tissues in both physiological and pathological states. While the paper presents interest for the community, the manuscript exhibits several significant limitations.

1. The primary limitation is the tool's focus predominantly on hematopoietic tissues. This restricts its applicability across a broader range of tissue types like the brain, lung, pancreas, or kidney, which are critical in many disease studies and fundamental biological research. Expanding its utility to include these and other tissues would make the tool more versatile and valuable across different fields of biomedical research. Consequently, it remains unclear how the Cell Marker Accordion surpasses ScType (<https://www.nature.com/articles/s41467-022-28803-w>), which supports annotations for more than 20 tissue types, automatically assigns cell types, and integrates multiple datasets and public sources.

This is an important point, raised by all reviewers. Following the reviewers' suggestions, the Cell Marker Accordion Database has been extended to include **23** source databases (previously 9), **212** human and **108** mouse tissues (previously 1 and 1), including brain, lung, pancreas, and kidney.

The new database is shown in **Fig.2A** in the revised manuscript:

A Databases of gene markers

Fig.2: The Cell Marker Accordion: a user-friendly platform for annotating and interpreting single-cell populations. A Workflow for building the Cell Marker Accordion database. Sources are ranked according to their initial number of markers. The resulting numbers of human and murine markers, cell types and tissues are reported. Mouse and human illustrations created in BioRender. Tebaldi, T. (2025) <https://BioRender.com/x09w717>.

In addition, the database for the annotation of disease-critical cells has also been expanded to include **196** human and **98** murine aberrant cell types, associated with **133** and **50** diseases, in **29** and **19** tissues, respectively.

The new database for annotation of disease-critical cells is shown in **Fig.5A** in the revised manuscript:

Fig.5: The Cell Marker Accordion identifies disease-critical cell types. A Workflow for building the Cell Marker Accordion Disease database. The resulting number of human and murine markers for aberrant cell types associated with various diseases from multiple tissues are reported. Mouse and human illustrations created in BioRender. Tebaldi, T. (2025) <https://BioRender.com/x09w717>.

The revised manuscript has been updated to describe these changes :

*“We built the Cell Marker Accordion database by integrating 23 marker gene databases and cell sorting marker sources (**Supplementary Table 1**), distinguishing positive from negative markers (**Fig.2A**). Standardization was achieved by mapping the initial cell type nomenclature to the Cell Ontology terms³⁹ and tissue names to the Uber-anatomy ontology (Uberon) terms⁴⁰. Next, via database integration, we obtained a comprehensive set of cell-type specific marker genes, human and murine, in hundreds of tissues.”*

*“We integrated marker genes associated with disease-critical cells found in multiple pathologies, including different tumor types affecting blood, brain, lung and pancreas (**Fig.5A**). To obtain a standardized and consistent vocabulary of pathologies, we mapped disease terms to the Disease Ontology (<https://disease-ontology.org/>)⁵²”*

*“After standardization and integration, Cell Marker Accordion includes a comprehensive collection of 15479 marker genes associated with 728 cell types in 212 human tissues, and 4118 marker genes associated with 493 cell types in 108 murine tissues (**Fig.2A**). Furthermore, the Cell Marker Accordion includes an extensive disease collection of 5876 genes associated with 196 aberrant cell types and 132 diseases in 29 human tissues, and 1567 genes associated with 98 aberrant cell types and 50 diseases in 19 murine tissues (**Fig.5A**).”*

The revised Cell Marker Accordion database surpasses the scType database³⁸ both in terms of sources (23 vs 2), tissues (270 vs 16), cell types (830 vs 193) and markers (19596 vs 2648). Unlike the scType database, the Cell Marker Accordion provides murine- and human-specific markers.

In the revised manuscript, we have further shown the improved performance of the Cell Marker Accordion over scType in 9 benchmark datasets (see also the answer to point 3).

2. While Cell Marker Accordion allows for identification of disease-critical single-cell populations in several cancer types, it is not clear how it surpasses tools like SCEVAN (<https://www.nature.com/articles/s41467-023-36790-9>), which supports more tissue types and is robustly validated in recent studies. A comparative analysis highlighting specific cases where the Cell Marker Accordion provides superior results in terms of accuracy, user experience, or unique features would greatly enhance its positioning within the research community.

This is another important point raised by all reviewers. During the revision, the Cell Marker Accordion database for annotating disease-critical cells has been expanded to include **196** human and **98** murine aberrant cell-types, associated with **133** and **50** diseases, in **29** and **19** tissues, respectively.

Following the reviewer's suggestion, we also included a comparative analysis between the Cell Marker Accordion and two other tools: SCEVAN and CopyKAT.

Both SCEVAN and CopyKAT identify tumor cells by calling copy number variations (CNV). For this reason, in liquid cancers such as leukemia, characterized by a minimal number of genomic alterations, these tools may not be suited. This limitation is also stated in the original SCEVAN article⁶⁴. On the other hand, we show that the Cell Marker Accordion can identify multiple leukemia aberrant cell types in AML patients (revised **Fig. 5**), and neoplastic plasma cells in Multiple Myeloma patients (revised **Supplementary Fig.6**). Additionally, the Cell Marker Accordion identifies and returns genes that have more impact on determining aberrant cell types.

In the revised manuscript, we directly compared the Cell Marker Accordion with SCEVAN and CopyKAT in two solid tumors: glioblastoma and lung adenocarcinoma. We show that the Cell Marker Accordion outperforms the other tools in terms of accuracy, and running time. Furthermore, while SCEVAN and CopyKAT can only distinguish malignant versus non-malignant cells based on inferred CNV signatures, we show that the Cell Marker Accordion can dissect aberrant sub-populations based on expression alterations.

These new benchmark results are shown in **Fig.6** in the revised manuscript.

Fig.6: The Cell Marker Accordion improves the identification of malignant cells in solid tumors. **A** Identification of malignant cells in glioblastoma patients. Left: original annotation²⁴. Right: Cell Marker Accordion annotation. **B** Comparison of annotation performances in identifying glioblastoma malignant cells, measured as the percentage of cells corresponding to the ground truth and the relative F1 scores.

C Comparison of annotation running times among tools **D** Identification of malignant and neoplastic endothelial cells in lung adenocarcinoma with the Disease Accordion. Left: original annotation²⁵. Right: Cell Marker Accordion annotation. **E** Comparison of annotation performances in identifying malignant cells (left panel) and endothelial cells with a neoplastic gene expression signature (right panel).

3. The manuscript lacks a comprehensive comparison with other state-of-the-art single-cell annotation tools. Such comparative analysis, including strengths, weaknesses, and performance benchmarks, would provide a clearer positioning of the tool in the landscape of single-cell analysis technologies. Detailing scenarios where it outperforms or falls short against other tools would help potential users make informed decisions.

This is another important issue, raised by all reviewers. In the revised manuscript, the Cell Marker Accordion has been extensively compared with state-of-the-art tools.

We conducted a performance benchmark study to compare the Cell Marker Accordion annotation performance against five other automatic tools based on markers (ScType³⁸, SCINA¹⁹, clustifyR¹², scCATCH⁴¹ and scSorter¹⁷), using multiple published single-cell datasets across different platforms and tissues: peripheral blood, bone marrow, immune system, pancreas, kidney, lung (see revised **Supplementary Table 4 and 5** for the list of datasets and benchmark settings).

To evaluate the annotation performance, in the revised manuscript, we introduced a new metric that considers the distance between the “predicted” and the “true” cell type using the Cell Ontology tree (see revised **Methods**), expressed as a percentage. In all cases, the Cell Marker Accordion couples higher annotation performances (+23% on average) with lower running time. Additionally, with respect to other tools, the Cell Marker Accordion provides unique output and visualizations to boost the interpretation of results.

These new benchmark results are shown in **Fig.3** in the revised manuscript.

Fig.3: The Cell Marker Accordion improves the annotation of cell types in multiple tissues from complex single-cell multiomics. Annotation of single-cell datasets and interpretation of the results with the Cell Marker Accordion, and performance comparison with other marker-based annotation tools. **A** Dataset of PBMC FACS sorted cells separately profiled with single-cell RNA-seq. 15 surface

antibodies were used to sort 10 different cell types, used as the ground truth. Populations identified by the Accordion are color-coded in the UMAP, with cluster numbers. **B** The Cell Marker Accordion annotation performance, measured as the average similarity between the identified cell types and the ground truth, is compared against other annotation tools. **C** Comparison of running times across annotation tools (time axis is log scaled). **D** Cell Marker Accordion interpretation of results: top three cell types achieving the highest impact score for each cell cluster (the winning cell type is highlighted). **E** Cell type annotation for cluster 5. Left: top three cell types, ordered according to their impact score, with corresponding percentages of cells in the cluster. Right: Cell Ontology tree of the top three cell types. **F** Top three marker genes with the highest impact score for each cell type color-coded as **E**. **G** Comparison of annotation performances between the Cell Marker Accordion and other tools in multiple single-cell datasets from different tissues.

Furthermore, we included an additional comparison on a spatial omics dataset: a recently published adult mouse brain MERFISH dataset, with a panel of 1122 genes⁴⁸. Also in the spatial scenario, the Cell Marker Accordion annotation performance improved by over 13% compared to the other tools. These results are shown in **Fig.4** in the revised manuscript.

Fig.4: The Cell Marker Accordion improves the annotation of brain cell types in spatial transcriptomics. **A** Spatial map and original annotation of a coronal section of an adult mouse brain, analyzed by MERFISH, based on a 1122 genes panel⁴⁸. Each dot corresponds to a cell, colored by cell type. Scale bar: 1 mm. **B** UMAP plot based on the transcriptional profile of each cell, with colors based on the annotation of the Cell Marker Accordion. **C** Spatial map with cells colored according to cell types as annotated by the Cell Marker Accordion. Scale bar: 1 mm. **D** Comparison across tools of annotation performances, measured as the similarities with the ground truth.

In the revised manuscript, we also provide a table where annotation tools used for the benchmark are compared in terms of required input/output, capabilities and features, and strengths/weaknesses (**Supplementary Table 2**).

It is important to point out that we considered only marker-based annotation tools in our benchmark. This issue, and the rationale for this choice, are discussed in the revised manuscript:

“Complementary to the Cell Marker Accordion, several annotation tools exist that do not directly rely on marker genes^{86,87}. These methods often rely on correlating reference expression data or transferring learned labels from other annotated single-cell datasets. However, these approaches necessitate high-quality and comprehensive reference datasets with annotated clusters for all relevant cell populations. Importantly, they can be susceptible to technical variations, such as differences in experimental platforms or sequencing strategies¹⁴. Given the current rate of evolution in single-cell and spatial omics technologies, this aspect cannot be underestimated. For this reason, a direct comparison between these approaches and the Cell Marker Accordion was not conducted in this study.”

Finally, we also included additional benchmark experiments in the context of detecting disease-associated cell types (see our answer to **major point 2**).

Minor points:

1. Although Figure 1 illustrates the variability in cell type annotations between CellMarker and PanglaoDB, ScTypeDB appears to effectively integrate and curate these databases, resulting in annotations that closely match those reported in the original article.

Please see our answer to **major point 1**, where we address the comparison between the revised Cell Marker Accordion and scTypeDB in terms of sources (23 vs 2), tissues (270 vs 16), cell types (830 vs 193) and markers (19596 vs 2648). In our answer to major point 3, we also describe the new benchmark experiments.

Finally, **Fig.1** has been updated in the revised manuscript to reflect the new data sources integrated in the Cell Marker Accordion (e.g., CellMarker2.0 was used, instead of CellMarker).

2. While the Cell Marker Accordion is marketed as an R package, it requires both a Python installation and a properly configured virtual environment for successful installation on Windows. This requirement should be explicitly stated and accompanied by comprehensive, step-by-step installation instructions, as the lack of such details currently makes the package non-functional on Windows systems without these prerequisites (based on tests conducted on three different machines).

We thank the reviewer for noting this problem. In the revision, we fixed this installation issue so that Python (or specific virtual environments) are not needed for a successful installation of the Cell Marker Accordion.

Reviewer #1 (Remarks on code availability):

While the Cell Marker Accordion is marketed as an R package, it requires both a Python installation and a properly configured virtual environment for successful installation on Windows. This requirement should be explicitly stated and accompanied by comprehensive, step-by-step installation instructions, as the lack of such details currently makes the package non-functional on Windows systems without these prerequisites (based on tests conducted on three different machines).

We thank the reviewer for noting this problem. In the revision, we fixed this installation issue so that Python (or specific virtual environments) are not needed for a successful installation of the Cell Marker Accordion.

Otherwise step-by-step guide was clear and easy to use.

Reviewer #2 (Remarks to the Author):

In this manuscript, the authors introduce the Cell Marker Accordion, a website and R package designed to improve the annotation and interpretation of scRNA-seq data. Traditional tools and databases often yield inconsistent cell type annotations due to variability in marker genes, particularly in pathological contexts. The Cell Marker Accordion addresses this important issue by integrating multiple gene marker databases and literatures, and weighting markers by specificity and evidence consistency. Validation across multiple hematopoietic cell lineage datasets demonstrated improved accuracy in cell type identification, including the ability to detect disease-critical cells in conditions like acute myeloid leukemia and multiple myeloma. This method holds the potential to become a powerful tool for studying hematopoietic cell lineage scRNA-seq datasets under both physiological and pathological conditions. However, there are some concerns in Cell Marker Accordion, particularly for its scope of application in diverse tissues and the lack of benchmarking with published methods. The comments are below.

Major:

1. The scope of application in diverse tissues is a serious concern for Cell Marker Accordion. Although the authors demonstrated the application of Cell Marker Accordion in several datasets in the manuscript, all the included datasets were generated from hematopoietic cell lineages. The performance of Cell Marker Accordion in scRNA-seq datasets generated from all other tissues/organs remains unknown. If the authors aimed to introduce Cell Marker Accordion as a general-purpose method for cell type annotation, its performance in diverse tissues should be evaluated. Otherwise, the authors at least need to clearly state in the abstract and discussion that Cell Marker Accordion is only tested on and applicable to hematopoietic cell lineages.

This is an important issue raised by all reviewers. During the revision, we expanded the Cell Marker Accordion so that it can be applied to multiple tissues as a general-purpose method for cell type annotation in both physiological and pathological contexts. Please see our response to **Reviewer 1, major point 1**, for a more detailed description.

2. Additional benchmarking with published single-cell annotation methods is necessary to evaluate the performance of Cell Marker Accordion. There are many published methods for

automatic annotating cell types in scRNA-seq data. However, this study only benchmarked Cell Marker Accordion with several cell marker gene datasets, but not benchmarked with bioinformatic methods for automatic annotating cell types. Please consider benchmarking Cell Marker Accordion with cell type annotation methods mentioned in two benchmarking studies: PMID: 31500660 and 33359678.

This is also an important issue raised by all reviewers. In the revised manuscript, we included multiple benchmark experiments against marker-based cell-type annotation methods, showing improved performance in multiple tissues and platforms. Please see our response to **Reviewer 1, major point 3**, for a detailed description of these benchmarks. We also included benchmark experiments for the detection of disease-critical cells. Please see our response to **Reviewer 1, major point 2**, for a detailed description.

3. The title “Interpreting single-cell messages in health and disease with the Cell Marker Accordion” does not summarize the work well. The meaning of “Interpreting single-cell messages in health and disease” is not clear to readers. The authors may change the title that appropriately reflect the functions and/or findings of Cell Marker Accordion.

Following this suggestion, in the revised manuscript, the title has been modified to: *“Cell Marker Accordion: interpretable single-cell and spatial omics annotation in health and disease”*.

4. The calculation of specificity and EC score should take positive and negative markers into the consideration. In the Methods section “Definition of integration scores for marker genes in the Cell Marker Accordion database”, there is no description of how to handle positive and negative markers. In particular, are there some genes listed as positive markers in one source and listed as negative markers in the other source? These genes should be considered conflicting evidence but not consistent evidence when calculating specificity and EC score.

We thank the reviewer for noticing this issue. Indeed, a few examples of conflicting evidence can be found when integrating the Cell Marker Accordion sources. this into consideration, in the revision, we changed the annotation algorithm so that the Evidence Consistency Score would indeed be penalized by conflicting evidence. Our solution is described in the revised **Methods**, in the sub-section “Implementation of the Cell Marker Accordion R package for automatic annotation”:

“Based on the input, ECs are computed for each gene G and cell type CT, considering positive and negative markers separately. In case of conflicting sources, i.e. genes listed as positive and negative markers for the same cell type depending on the source, the ECs is penalized: ECs of negative occurrences are subtracted from ECs of positive occurrences to obtain the final ECs, whose sign will determine whether the marker is classified as negative (<0), positive (>0) or not considered (=0), and the final ECs is defined as the absolute value.”

Minor:

1. In the Results section “Widespread heterogeneity across annotation sources leads to inconsistent cell type annotation”, it is not clear how Jaccard similarity among marker gene databases are calculated. It is suggested to add some paragraphs in Methods to describe

what cell types and what marker genes were used to calculate Jaccard similarity, and please provide the marker genes from datasets as a supplementary file if possible.

The overlap analysis and Fig.1 have been updated considering the new sources used by the Cell Marker Accordion. In the revised manuscript, we included a new **Methods** sub-section, titled: "Overlap among published marker gene databases":

"To quantify the overlap of marker genes between two marker gene databases (Fig.1C), we calculated the Jaccard pairwise similarity values. This was determined as the average Jaccard similarity of marker genes associated with common cell types present in both databases"

Finally, as suggested by the reviewer, all markers from each source can be retrieved with the Cell Marker Accordion shiny app: we provide the web server link and the possibility for any user to download and run the source code, available on GitHub. The full database is also available in the data folder of the Cell Marker Accordion R package. In the revised manuscript, we included the list of data sources with all the associated references as a supplementary table (**Supplementary Table 1**).

2. Figure 4A, it is not clear what marker genes and literatures for AML were used by Cell Marker Accordion. Please cite the relevant literatures and report the markers in a supplementary file and/or on shiny App.

In the revision, we updated the Cell Marker Accordion Database, extending the scope to multiple tissues and diseases. All disease-associated markers, together with references, are available in the database and can be downloaded from the shiny app or the Github source code. The full database is also available in the data folder of the Cell Marker Accordion R package. In the revised manuscript, we included the list of data sources with all the associated references as a supplementary table (**Supplementary Table 1**).

3. Supplementary Figure 5: it is better to add a UMAP plot showing the cell cluster labels, which are shown in Supplementary Figure 5E but not in UMAP.

According to the reviewer's suggestion, the figure (now **Supplementary Figure 6**) has been substantially revised, and cluster labels were included in the UMAP.

4. Sentence "By single-cell mutation calling on reads mapping to the U2AF1 locus (see Methods)", there is no methods for mutation calling in the manuscript.

The **Method** section of the revised manuscript contains a sub-section titled: "MDS single-cell dataset", with a description of the mutation calling procedure:

"Cell-variant assignment, based on U2AF1 S34F mutation calling in the 21:43104346-43104346 locus, was performed with VarTrix v1.1.19 (<https://github.com/10XGenomics/vartrix>)"

5. Figure 5B, it is better to show the cell percentage in individual patients from two groups (5 from SF_WT and 3 from U2AF1), because the merged percentage may be misleading when there is strong intra-group variation. Figure 5C-D, using two contrasting colors would be better to distinguish the two groups.

We agree with this point: in the revised **Fig.7B**, cell percentages of individual patients are now visualized.

Fig.7: The Cell Marker Accordion identifies cell type alterations in splicing factor mutant cells from patients with myelodysplastic syndromes. A Cell Marker Accordion cell type annotation of MDS patients with and without U2AF1 S34F mutation. **B** Changes in the abundance of hematopoietic cell types among conditions. Orange bars represent patients with U2AF1 S34F mutations, and grey bars represent patients without splicing factor mutations. Compositional analysis was performed with the scCODA77 python package based on Bayesian models. Credible and significant results are highlighted as blue bars, using an FDR threshold of 0.1.

6. Similar to the comment 5 above, Figure 6C,D,G,H, it is better to show the cell percentage in individual mice from the two groups in each study.

In line with the previous comment, cell percentages of individual mice are now visualized in the revised **Fig.8 C and G**. We kept the original visualization for cell cycle stage panels, where the stacked barplot visualization is less compatible with the display of single sample points.

7. Please add a Data Availability statement for the data generated in this study.

The **Data Availability** and **Code Availability** sections were added to the revised manuscript.

Reviewer #2 (Remarks on code availability):

Please find the attached code review report to improve the R package.

We thank the reviewer for this additional feedback. The Cell Marker Accordion R package has been improved to address all the raised points. In particular:

- 1) the problem with dependencies was fixed. Now all dependencies are automatically installed.

- 2) Errors and warnings occurring during the installation and loading of the packages have been fixed (points a-d).

Reviewer #3 (Remarks to the Author):

Gene marker-based methods are commonly used to annotate cell types in scRNA-seq data. However, depending on the list of gene markers assigned to each cell type and the gene marker databases used, the results can be highly variable even if the same annotation algorithm is applied. In this study, Busarello et al. have addressed this important problem by developing a novel method called Accordion, which takes advantage of using multiple gene marker databases to improve cell type annotations results. This represents an important step forward in the field.

Major Comments:

Comment 1: This study has focused on immune cells and hematopoietic cell types. It is well known that inconsistencies in cell type annotation results due to gene markers are a general problem that applies to many different tissue types, not just immune cells. While single-cell RNA-seq data and relevant gene markers are available for a wide range of cell types and tissues, my question is: "why hasn't Accordion been developed for a broader range of cell types and tissues? What factors might have limited the implementation of Accordion for other cell types, and its utility for broader and more general applications? Are there any plans to extend this study to non immune related cell types and diseases?"

This is an important issue raised by all reviewers. During the revision, we expanded the Cell Marker Accordion so that it can be applied to multiple tissues as a general-purpose method for cell type annotation in both physiological and pathological contexts. Please see our response to **Reviewer 1, major point 1**, for a more detailed description.

Comment 2: When considering the improvement in matching with FACS populations, the results have substantially improved for PBMCs from Zheng et al. 2017 with improvement in percentage of match > 20% (Figure 3B); however, the improvements are more subtle for the bone marrow data (Triana 2021) and the multi-study scRNA-seq data (comprising a census of immune cells, bone marrow, and umbilical cord blood datasets) (Improvement in percentage of match is around 5-8%) (Supplementary Figures 3A and B). It is important to discuss where and under what conditions this approach would work best and realize its full potential, and what factors are necessary to achieve substantial improvements in the results.

This is also an important issue raised by all reviewers. In the revised manuscript, we included multiple benchmark experiments against marker-based cell-type annotation methods, showing improved performance in multiple tissues and platforms. To evaluate the annotation performance, in the revised manuscript, we introduced a new metric that considers the distance between the "predicted" and the "true" cell type using the Cell Ontology graph, expressed as a percentage (see revised **Methods**). In all cases, the Cell Marker Accordion achieves higher annotation performances (+23% on average). Please see our response to **Reviewer 1, major point 3**, for a detailed description of these benchmarks.

Comment 3: I appreciate that the authors have tested Accordion across several datasets and demonstrated its utility with a wide range of applications (healthy individuals, disease-relevant cells; human and mouse data). However, for the case studies presented in **Figures 4-6**, it is important to see how other single annotation methods perform and how much improvement (in terms of biological discoveries) is gained by using Accordion compared to single databases. Are the other single database methods capable of making similar discoveries, or do they fail? This comparison will help us understand the extent of the improvement gained from Accordion compared to single-database approaches. Is it a subtle enhancement, or is it substantial?

This comment is partially linked to the previous one and equally important. Apart from extensive benchmarks in multiple physiological tissues, in the revised manuscript we also included benchmark experiments for the detection of disease-critical cells, and compared the Cell Marker Accordion with other tools. Please see our response to **Reviewer 1, major point 2**, for a detailed description.

Comment 4: The incorrect or inconsistent cell type annotation issue presented in Figure 1 can be influenced by the chosen cell type annotation algorithm (in this case, scType). It would be interesting to test this across multiple annotation algorithms (e.g., SCINA, clustifyR, scCATCH, scSorter) and present the level of similarity (measured by Jaccard) across single databases for various annotation algorithms, not just scType.

In the revised manuscript, **Fig.1** has been updated to include new data sources used in the Cell Marker Accordion.

Fig.1: Heterogeneity in marker gene databases leads to inconsistent single-cell annotations. A Cell type identification by automatic annotation with ScType³⁸ in the Oetjen et al., 2018³⁰ bone marrow dataset, using markers from CellMarker2.0 (left) and PanglaoDB (right) as input. **B** Overlap between marker genes from CellMarker2.0 (y-axis) and PanglaoDB (x-axis). The dot color represents the Jaccard similarity index, and the dot size indicates the number of common markers in each cell type pair. **C** Comparison of cell type markers in published databases. The numbers indicate the average Jaccard similarity index between each database pair, calculated using all common cell types.

Following the suggestion of the reviewer, benchmark analysis comparing the Cell Marker Accordion with other tools (including scType, SCINA, clustifyR, scCATCH, scSorter) have been performed and reported in revised **Fig. 3** and **Fig.4**

Minor Comments:

Page 9, in the sentence "which are one of the the key factors contributing". "The" is repeated. Please correct it.

We thank the reviewer. This error has been fixed.

On Page 19, for the following sentence: "Then, both specificity and EC score are log transformed and multiplied to obtain a comprehensive weight for each marker gene in each cell type."

1 - Could you provide a more complete justification for measuring the weight in this way? What is the range of values for the calculated weights?

2 - Is "i." after the above sentence a typo?

1. In the revised manuscript, additional details on the algorithm were included in the **Methods** section, and in **Supplementary Fig. 1**. In particular:

"ECs, ranging from 1 to 23 in the current Accordion database for both positive and negative markers, are log10 transformed to obtain regularized evidence consistency scores (ECs_reg) (Supplementary Fig. 1). Next, the reciprocal of SPs is reverse scaled, separately for positive and negative marker. The result is then scaled to the same range of the ECs and log10 transformed, obtaining the regularized specificity score (SPs_reg). This procedure gives the same overall weight to the two scores. The resulting SPs_reg and ECs_reg are multiplied to obtain a final weight for each marker gene in each cell type. This final weight ranges from 1 to a maximum that depends on the marker database (5.6 in the current Accordion database, a magnitude similar to scaled gene expression levels)."

2. Thank you. The typo has been fixed.

In the last paragraph of Page 3, it is mentioned: "... with an average Jaccard similarity of 0.04 and a maximum of 0.12." However, in Figure 1C, it appears that the Jaccard similarity is around 0.5. These two values seem inconsistent.

Fig. 1C has been updated to obtain a clearer visualization of the Jaccard similarity values obtained from the pair-wise comparison of databases (see our answer to **comment 4**, where revised **Fig.1** is included).

I suggest that the formulas presented in Supplementary Figure 1 be included in the Methods section (first half; Page 18), so that the reader can gain a more complete understanding of the details related to the annotation and scoring workflow.

Following the reviewer's suggestion, the revised **Methods** include the formulas of the main steps of the Cell Marker Accordion algorithm.

Reviewer #3 (Remarks on code availability):

The README file on the GitHub page provides sufficient detail to run the code. I was not successful in installing cellmarkeraccordion on my machine due to discrepancies with the dependencies, but that may be an issue on my end rather than a problem with the package.

We thank the reviewer for this feedback. All the installation problems due to errors with dependencies have been fixed.

We are happy to submit the revised manuscript for submission NCOMMS-24-44973.

We have now successfully addressed all the reviewers' additional points. We have invested significant effort in enhancing not only the manuscript and related materials, but also:

1. The Cell Marker Accordion web Shiny app (<https://rdds.it/CellMarkerAccordion/>), with notable improvements to usability.
2. The GitHub page and README for the Shiny-Cell Marker Accordion (https://github.com/TebaldiLab/shiny_cellmarkeraccordion), providing clearer documentation.
3. The GitHub page and README for the Cell Marker Accordion R package (<https://github.com/TebaldiLab/cellmarkeraccordion>), with improved instructions and examples.

The purpose of these modifications was to improve user experience by incorporating supplementary explanations, illustrative examples, and convenient downloadable links.

We believe these revisions have significantly strengthened our tool, and we sincerely appreciate the reviewers' insightful comments.

Reviewer's Comments:

Reviewer #1 (Remarks to the Author)

The authors made a thorough revision of the manuscript. I am generally pleased with the improvements, particularly regarding the installation issues and usability of the GitHub repository. The Cell Marker Accordion now demonstrates broader applicability across tissues and improved benchmarking. However, I have several remaining concerns:

1. While the integrated and curated database represents a major contribution to the field, I didn't find a simple way to download it. The text mentions the database is available through the Shiny app, R package data folder, and GitHub, but none of these provide straightforward access to the raw data. Given that many researchers may want to integrate these valuable marker genes with their own analyses or computational pipelines without needing to use the full R package, I strongly recommend providing direct downloads of the curated database as simple Excel/CSV files. These should be separated into normal human tissues, normal mouse tissues, human disease states, and mouse disease states. Such accessibility would significantly enhance the utility of this important resource for the scientific community.

Following the reviewer's suggestion, the Cell Marker Accordion database is now also provided as an Excel file with 4 different worksheets ("Human_healthy", "Mouse_healthy", "Human_disease",

“Mouse_disease”) and a README describing all tables. The file can now be downloaded from multiple locations:

- 1) Supplementary Table 8 in the revised manuscript
- 2) The online Cell Marker Accordion shiny web app (<https://rdds.it/CellMarkerAccordion/>), by clicking the “Download” button in the sidebar of the web page

- 3) The GitHub repository of the shiny app (https://github.com/TebaldiLab/shiny_cellmarkeraccordion)

- 4) The GitHub repository of the R package (<https://github.com/TebaldiLab/cellmarkeraccordion>)

Access and download the Accordion database

To access the *healthy* Accordion database run:

```
data(accordion_marker)
```

To access the *disease* Accordion database run:

```
data(disease_accordion_marker)
```

To download the Accordion database as an Excel file click the Download button in the Cell Marker Accordion Shiny app available at: <https://rdds.it/CellMarkerAccordion/>.

 Download the
Cell Marker Accordion database

Alternatively, download the "TheCellMarkerAccordion_database_v0.9.7.xlsx" file from the Shiny app's GitHub repository stored in the "data" folder: Download TheCellMarkerAccordion_database from Shiny app repo.

Additionally, instructions to download and load in R the Accordion Cell Marker Database were included in the GitHub repository of the shiny web app and the R package.

2. Disease State Generalizability: While the authors expanded the disease database significantly, I notice that the validation focuses heavily on certain cancer types. To claim broad applicability across malignancies, the authors should either: a) Demonstrate validation across more diverse cancer types and tissues, or b) More carefully scope the claims about malignancy detection to the specific cancer types they have validated

As helpfully pointed out by the reviewer, we have more carefully scoped our claims about malignancy detection in the revised manuscript, incorporating the following limitation statement:

While the Cell Marker Accordion's customizability suggests broad applicability in cancer and other diseases, our current benchmark, focused on two liquid and two solid tumors, demonstrates improved performance within these specific contexts. We plan to expand this benchmark in the future as the availability of high-quality, ground truth datasets increases.

Minor Points:

1. Consider adding a brief "Data Access Guide" section to the manuscript describing the different ways researchers can access and utilize the curated database.

The following "Data Access Guide" section has been included in the revised manuscript, in the "Data availability" section.

Data Access Guide

The Cell Marker Accordion curated database is available as an Excel file (Supplementary Table 8). Additionally, it is available in the “data” folder of the GitHub repository of the Cell Marker Accordion Shiny web app (https://github.com/TebaldiLab/shiny_cellmarkeraccordion). The curated Cell Marker Accordion database can also be downloaded from the online Shiny web app (<https://rdds.it/CellMarkerAccordion/>) by clicking the “Download” button in the sidebar. The GitHub repositories for both the Shiny web app and the R package (<https://github.com/TebaldiLab/cellmarkeraccordion>) contain simple instructions on how to access and download the Cell Marker Accordion database.

The Cell Marker Accordion database can be easily explored or customized using the Shiny web app (<https://rdds.it/CellMarkerAccordion/>), which allows to: a) search and download lists of marker genes associated with input cell types across different tissues in health and disease; b) search and download lists of cell types associated with input marker genes across different tissues in health and disease; c) integrate custom set of marker genes with the Cell Marker Accordion database; d) perform cell type marker enrichment analysis across tissues in health and disease. Additionally, users can also browse hierarchies of cell types following the Cell Ontology structure in order to obtain the desired level of specificity in the markers and rank and select marker genes by their evidence consistency and specificity scores.

Demo examples are provided for guidance. All resulting tables can be easily explored and downloaded. A tutorial to the web app’s functionalities, along with instructions for local installation, is available on the GitHub page of the Shiny app: https://github.com/TebaldiLab/shiny_cellmarkeraccordion.

2. In the GitHub repository, include example scripts showing how users can integrate their own marker genes with the database.

In accordance with the reviewer's constructive feedback, a new function, "marker_database_integration", has been added to the R package, enabling quick and easy integration of custom gene sets with the Cell Marker Accordion database. A new tutorial ("Integrate custom sets of markers with the Cell Marker Accordion database") has been added to the README on the GitHub page of the R package. The first part is dedicated to the description of the parameters to perform the integration. For a more effective explanation, we included an example:

Integrate custom set of markers with the Accordion database

The `cellmarkeraccordion` package includes the `marker_database_integration` function, which allows users to integrate a custom set of marker genes into the Accordion database—either for healthy or disease conditions.

Usage

Set the `database` parameter to either:

- "healthy" → Integrate with the healthy Accordion database
- "disease" → Integrate with the disease Accordion database

Input Requirements

The function requires a marker gene table with at least two columns:

- "cell_type" – Specifies the cell type
- "marker" – Lists the marker genes

To ensure proper integration, cell types nomenclature should be standardized:

- Healthy database → Use Cell Ontology
- Disease database → Use NCI Thesaurus

If non-standardized cell types are provided, they will be added as "new" cell types in the database.

Running the Integration

Load custom set of marker genes:

```
load(system.file("extdata", "custom_markers_to_integrate.rda", package = "cellmarkeraccordion"))
head(custom_markers_to_integrate, 10)
```

To integrate the custom table with the healthy Accordion database, use:

```
database_integrated<-marker_database_integration(marker_table = custom_markers_to_integrate,
  database = "healthy",
  species_column = "species",
  disease_column = "disease",
  tissue_column = "Uberon_tissue",
  celltype_column = "CL_celltype",
  marker_column = "marker",
  marker_type_column = "marker_type",
  resource_column = "resource")
```

To perform automatic cell type annotation using the previously integrated marker database, pass the output table from `marker_database_integration` to the `database` parameter of the `accordion` function (or `accordion_disease` if the integration has been performed with the disease database of the Cell Marker Accordion).

Additionally, the Shiny web app now includes a dedicated page for seamless custom marker integration, which does not require computational skills. A demo example is also provided for guidance.

3. For transparency, please include in the supplementary materials a complete list of all sources used for database curation, including how each source was processed and integrated.

Additional information on how each source was processed has been included in the revised Supplementary Table 1. Moreover, the downloadable Cell Marker Accordion database includes columns that display the original cell type, tissue, and disease information, along with their standardized versions after integration.

(Remarks on code availability)

The code was easy to use, but I didn't find an easy way to access the database.

This issue has been addressed, please see the answer to point 1 for further details.

Reviewer #2 (Remarks to the Author)

During the revision the authors made a significant improvement on Cell Marker Accordion to extend its scope from hematopoietic cell lineage to diverse tissues. In addition, Cell Marker Accordion was benchmarked with several single-cell annotation methods and demonstrated its strength. However, there are still some minor issues:

1. The authors mapped cell types to Cell Ontology terms and tissues to Uberon terms. This is a good standardization step, but no details were provided. A table of the mapping relationship may be useful to show the relevant information. Please consider adding this information to the webserver (better for webserver users) or as a supplementary table.

This is a good point. The “complete” table of the Cell Marker Accordion database on the web server, as well as the downloadable version, includes columns that display the original cell type,

tissue, and disease information, along with their standardized versions after integration (see the example screenshot below). Please see also our response to Reviewer 1, Point 1, where we describe how we implemented multiple options for downloading the Cell Marker Accordion database.

species	original_tissue	Uberon_tissue	Uberon_ID	tissue_definition	original_celltype	CL_celltype	CL_ID	cell_definition	marker	marker_type
Human	BRAIN	brain	UBERON:0000955	The brain is the center	Cajal-Retzius cell	Cajal-Retzius cell	CL:0000695	A neuron of the huma	CALB2	positive
Human	Brain	brain	UBERON:0000955	The brain is the center	Cajal-Retzius cells	Cajal-Retzius cell	CL:0000695	A neuron of the huma	CALB2	positive
Human	Brain	brain	UBERON:0000955	The brain is the center	Interneurons	interneuron	CL:0000099	Most generally any nei	CALB2	positive
Human	brain	brain	UBERON:0000955	The brain is the center	neural cell	neural cell	CL:0002319	A cell that is part of th	CALB2	positive
Human	Brain	brain	UBERON:0000955	The brain is the center	Neuroblast (sensu Vertebrata)	neuroblast (sensu Vertebrata)	CL:0000031	A cell that will develop	CALB2	positive
Human	Brain	brain	UBERON:0000955	The brain is the center	Neuron	neuron	CL:0000540	The basic cellular unit	CALB2	positive
Human	Brain	brain	UBERON:0000955	The brain is the center	Neurons	neuron	CL:0000540	The basic cellular unit	CALB2	positive
Human	Brain	brain	UBERON:0000955	The brain is the center	Trigeminal neurons	trigeminal neuron	CL:4023169	A neuron that is respo	CALCA	positive

2. The webserver requires improvements and bug fixes in the following areas:

Search by Tissue and Cell Types – When using this feature, I randomly selected various tissues and cell types. However, after repeating this process approximately five times, I was disconnected from the server.

Thank you for noticing this. In this revision, we addressed issues with the web server, with notable improvements to usability. Since we cannot totally exclude the possibility of server downtime, we also included clear instructions for local installation of the shiny Cell Marker Accordion in the README of the GitHub page (https://github.com/TebaldiLab/shiny_cellmarkeraccordion).

Local installation

Prerequisites

Before installing the Cell Marker Accordion shiny web app, ensure you have the following:

- R (Version 4.0 or higher) – Download from CRAN (<https://cran.r-project.org/>).
- RStudio (Recommended) – Download from RStudio (<https://posit.co/downloads/>).
- Python (Version 3.6 or higher) – Download from Python.org (<https://www.python.org/>).

1. Clone the Repository:

```
git clone https://github.com/TebaldiLab/shiny_cellmarkeraccordion.git
```

If Git is not installed, you can:

- Download and install Git from <https://git-scm.com/>.
- Alternatively, download the ZIP file and extract it.

2. Run the Installation Script:

Open R or RStudio, navigate to the cloned repository folder, and execute:

```
setwd("path/to/shiny_cellmarkeraccordion")
source("install_dependencies.R")
```

This script will:

- Install required R packages
- Set up the necessary Python environment
- Install Python dependencies

3. Run the Shiny application

```
library(shiny)
runApp()
```

Cell Type Annotation – I uploaded the required differentially expressed gene table, but encountered the error message during annotation: "An error has occurred. Check your logs or contact the app author for clarification." This issue persists regardless of the parameters set.

Thank you for noticing this. These errors have been reviewed and fixed. We also included a tutorial, with a demo file, on how to perform cell type marker enrichment analysis on the Cell Marker Accordion web server.

-  Homepage
-  Search by tissue and cell types
-  Search by marker genes
-  Custom markers integration

Marker enrichment analysis

 Download the Cell Marker Accordion database

Perform cell type marker enrichment analysis across tissues in health and disease

 Info

Load file

 No file selected  InputFile

Specify the number of positive genes to keep for each cluster <input checked="" type="radio"/> ALL Type in number of positive genes <input style="width: 100%;" type="text"/>	Specify the number of negative genes to keep for each cluster <input checked="" type="radio"/> ALL Type in number of negative genes <input style="width: 100%;" type="text"/>
--	--

 Load demo example

Output table generated by the FindAllMarkers function in the Seurat package

Interpreting single-cell messages in health and disease with the Cell Marker Accordion

Manuscript ID: NCOMMS-24-44973

Comments on testing code: R package cellmarkeraccordion:

There are multiple warnings and errors when using the package, and this reviewer was not able to complete running the tutorial available at <https://github.com/TebaldiLab/cellmarkeraccordion/blob/master/README.md>. Please consider making a significant improvement of the package.

Specific comments:

1. For dependency on BioConductor packages 'ontoProc', 'ontologyPlot', please guide the users to install these BioConductor packages before installing cellmarkeraccordion, because they will not be automatically installed.

2. Please address the following error and warnings when installing and loading cellmarkeraccordion R package:

a). When installing and loading the package on Windows system, there always got warnings

"WARNING: Retrying (Retry(total=4, connect=None, read=None, redirect=None, status=None)) after connection broken by 'SSLError("Can't connect to HTTPS URL because the SSL module is not available.")': /simple/owlready2/"

b). When loading the package, there were multiple warnings like this:

"replacing previous import 'data.table::transpose' by 'purrr::transpose' when loading 'cellmarkeraccordion' "

c). When running the code

```
"data <- accordion(data, include_detailed_annotation_info = T, plot=T, max_n_marker = 30,
annotation_resolution = "cluster", annotation_name = "accordion_pbmc", allow_unknown = F)"
```

There got the following error:

"Error in `[.data.table` (top_celltypes, , `:=`(win_ct, .SD[1]), by = "group") : Supplied 5 items to be assigned to 51 items of column 'win_ct'. If you wish to 'recycle' the RHS please use rep() to make this intent clear to readers of your code."

d). No results were generated in "Enhanced biological interpretation of results" with the error in point c).

e). Please test running the code before putting it into the README.md. There are some errors in the README.md file. For example, "accordion_cellcycle" should be corrected to "accordion_cell_cycle"; "load(bone_marrow_data)" should be corrected to "data(bone_marrow_data)".

During the revision the authors made a significant improvement on Cell Marker Accordion to extend its scope from hematopoietic cell lineage to diverse tissues. In addition, Cell Marker Accordion was benchmarked with several single-cell annotation methods and demonstrated its strength. However, there are still some minor issues:

1. The authors mapped cell types to Cell Ontology terms and tissues to Uberon terms. This is a good standardization step, but no details were provided. A table of the mapping relationship may be useful to show the relevant information. Please consider adding this information to the webserver (better for webserver users) or as a supplementary table.

2. The webserver requires improvements and bug fixes in the following areas:

Search by Tissue and Cell Types – When using this feature, I randomly selected various tissues and cell types. However, after repeating this process approximately five times, I was disconnected from the server.

Cell Type Annotation – I uploaded the required differentially expressed gene table, but encountered the error message during annotation: "An error has occurred. Check your logs or contact the app author for clarification." This issue persists regardless of the parameters set.

- Homepage
- Search by tissue and cell types
- Search by marker genes
- Cell types annotation

Load file to annotate
gptd_20180501...
Upload file

Select species:
 Human Mouse

Condition:
healthy

Tissue:
abdominal system, aorta of leg, anterior cingulate cortex, arachnoid of lymph node, auditory cortex, basal forebrain, biliary ductule, blood, blood brain barrier, blood platelet

Tissue covare:

How many positive markers you want for each cluster?
 ALL

How many negative markers you want for each cluster?
 ALL

Type in number of positive markers
10

Type in number of negative markers
10

Add value

Filters View

IC_score
 >=1
 >=2
 >=3
 >=4
 >=5
 >=6
 >=7

specificity_score
 >=0
 >=0.25
 >=0.5
 =1

Error: An error has occurred. Check your logs or contact the app author for clarification.

Click to annotate